# Staphylococcal phages and pathogenicity islands drive plasmid evolution

Suzanne Humphrey [1], Álvaro San Millán [2], Macarena Toll-Riera [3], John Connolly[1], Alejandra Flor-Duro[4], John Chen[5], Carles Ubeda [4,6], R. Craig MacLean [7] & José R. Penadés [1,8,9 ✉]

Conjugation has classically been considered the main mechanism driving plasmid transfer in nature. Yet bacteria frequently carry so-called non-transmissible plasmids, raising questions about how these plasmids spread. Interestingly, the size of many mobilisable and non-transmissible plasmids coincides with the average size of phages (~40 kb) or that of a family of pathogenicity islands, the phage-inducible chromosomal islands (PICIs, ~11 kb). Here, we show that phages and PICIs from *Staphylococcus aureus* can mediate intra- and inter-species plasmid transfer via generalised transduction, potentially contributing to non-transmissible plasmid spread in nature. Further, staphylococcal PICIs enhance plasmid packaging efficiency, and phages and PICIs exert selective pressures on plasmids via the physical capacity of their capsids, explaining the bimodal size distribution observed for non-conjugative plasmids. Our results highlight that transducing agents (phages, PICIs) have important roles in bacterial plasmid evolution and, potentially, in antimicrobial resistance transmission.

[1] Institute of Infection, Immunity and Inflammation, University of Glasgow, Glasgow G12 8TA, UK. [2] Centro Nacional de Biotecnología–CSIC, Madrid, Spain. [3] Institute of Integrative Biology, ETH Zurich, Zurich, Switzerland. [4] Fundación para el Fomento de la Investigación Sanitaria y Biomédica de la Comunitat Valenciana-FISABIO, 46020 Valencia, Spain. [5] Department of Microbiology and Immunology, Yong Loo Lin School of Medicine, National University of Singapore, 5 Science Drive 2, Singapore, Singapore. [6] Centers of Biomedical Research Network (CIBER) in Epidemiology and Public Health, Madrid, Spain. [7] Department of Zoology, University of Oxford, Oxford OX1 3SZ, UK. [8] Departamento de Ciencias Biomédicas, Facultad de Ciencias de la Salud, Universidad CEU Cardenal Herrera, Valencia 46113, Spain. [9] MRC Centre for Molecular Bacteriology and Infection, Imperial College London, London SW7 2AZ, UK. ✉email: j.penades@imperial.ac.uk

Bacteria account for around half of the cells of an average human body and are key determinants of health and disease. Despite decades of research, pertinent and fundamental questions about how bacteria evolve, as well as how and why multi-resistant and virulent clones emerge, continue to be unexplained. Antibiotic-resistant infections are proposed to be the major cause of death in 2050[1], creating an urgency to understand the processes driving bacterial evolution and the acquisition of virulence and antibiotic resistance genes (ARGs). Since ARGs are mostly carried by plasmids, we must decipher how these mobile genetic elements (MGEs) spread these genes in natural populations of bacteria, promoting the emergence of multi-resistant clones.

It has been classically assumed that conjugation is the main mechanism driving plasmid transfer, and drugs blocking this process are currently proposed as an alternative strategy to combat antibiotic resistance[2,3]. Moreover, classical mathematical models of plasmid population dynamics suggest that conjugation is required for the long-term survival of plasmids in bacterial populations[4,5]. Notably, despite the ability of many plasmids to self-transfer and their tendency to confer multi-drug resistances upon their host cell, frequently bacterial pathogens do not carry these elements, implying an inherent cost associated with their carriage[6]. Some plasmids have evolved strategies to reduce their cost by exploiting the transfer machinery encoded by conjugative plasmids: these are known as 'mobilisable' plasmids. Some mobilisable plasmids encode their own distinct relaxase (Mob) proteins, which recognise the cognate origin of transfer (*ori*T) present on the mobilisable plasmid, and essentially hijack the conjugative plasmid-encoded mating channels[7,8]. Other mobilisable plasmids carry only an *ori*T sequence in their genomes with transfer dependent on a compatible *trans*-acting relaxase being provided by a co-resident conjugative plasmid[8,9]. Critically, the reliance of these small mobilisable plasmids on a helper plasmid to facilitate their transfer highlights the complex interplay occurring between distinct MGEs within the bacterial cell.

A high throughput analysis of plasmid sequences has classified approximately half of all sequenced bacterial plasmids as 'non-transmissible', thus rendering their mechanism of transfer unknown[10]. The fact that 'non-transmissible' plasmids are widely distributed among the bacterial genomes implies the existence of an unrecognised mechanism of transfer permitting their spread and persistence in nature. Since non-transmissible plasmids are widely present in both naturally competent and non-competent bacteria, we hypothesised that transduction, but not transformation, is likely to be the mechanism used by non-transmissible plasmids to spread in nature. What evidence is there to support this idea?

While most conjugative plasmids have large genomes (usually > 60 kb)[10], the size distribution of the non-transmissible plasmids shows two interesting peaks: one of the size of average phages (~40 kb) and another approximately the size of phage inducible chromosomal islands (PICIs; ~11 kb)[10]. Intriguingly, the size distribution of the mobilisable plasmids also shows a predominant peak compatible with the size of the phage genomes[10].

PICIs are widespread phage satellites, present both in Gram-positive and Gram-negative bacteria[11,12], whose induction and transfer is promoted by helper phages[13,14]. Once induced, PICIs block phage reproduction and hijack the phage machinery for their own benefit, promoting the formation of small PICI-sized capsids[15], into which the complete PICI genome, but not the phage genome, can be packaged. This process of remodelling phage capsids to produce small PICI capsids has been particularly well characterised in the case of the staphylococcal PICI family, the *Staphylococcus aureus* Pathogenicity Islands (SaPIs), with the prototypical SaPI1 island producing capsids with around 33% of

the DNA packaging capacity of its 80α helper phage via the activity of two island-encoded capsid morphogenesis genes, *cpmAB*[16].

Importantly, both phages and PICIs engage in generalised transduction (GT)[17], suggesting that these elements and their interactions may have a huge impact on the biology and mobility of plasmids. GT is the process by which phages or PICIs that use the headful mechanism of packaging mobilise either chromosomal or plasmid DNA from one bacterium to another[18]. The process is initiated by either the phage- or PICI-encoded small terminase subunit (TerS) which occasionally recognises, with low frequency, *pac* site homologues (also called pseudo-*pac* sites) in host chromosomal or plasmid DNA, initiating their packaging into the phage or PICI capsid to form transducing particles. Since transducing particles can mobilise DNA of commensurate size with the phage or PICI genome (packaged in phage or PICI capsids, respectively), the size distribution observed with the non-transmissible and mobilisable plasmids suggests they may be transferred by transduction, using either phage-sized or PICI-sized capsids. If correct, we will establish a mechanism by which non-transmissible plasmids are transferred in nature and will identify additional strategies promoting the spread of the mobilisable plasmids. Moreover, these findings will also imply that phages and PICIs impose a trade-off between plasmid size and mobility, because plasmids that become very large by the acquisition of novel genes may no longer be mobilisable by the same transducing agents. Since non-transmissible and mobilisable plasmids usually encode antibiotic resistance genes, our results suggest that PICIs and phages could play a relevant role in the emergence of multi-resistant clones.

Here, we confirm these hypotheses and show that phages and PICIs drive plasmid evolution and mediate intra- and inter-species plasmid transfer via generalised transduction.

## Results

**Staphylococcal plasmids exhibit bimodal size distribution**. We selected *Staphylococcus aureus* as the experimental model in which to test the hypothesis that phage- and PICI-mediated GT impacts plasmid evolution and transfer because this species not only affords plasmids of different sizes and functions, but also features a number of transducing phages and the best-characterised members of the PICI family, the SaPIs[19]. We first analysed whether the size of the *S. aureus* plasmids has a similar distribution to that observed by Smillie and coworkers[10]. 295 *S. aureus* genomes, carrying 819 prophages and 243 plasmids, were acquired from the NCBI Database. Analysis of the size distribution of staphylococcal prophages showed a peak at approximately 44 kb (Fig. S1a). Consistent with the previous report, size distribution analysis of the 243 plasmids revealed two peaks within the SaPI-transducible (up to ~15 kb) and phage-transducible (up to ~45 kb) size ranges (Fig. S1b), confirming the suitability of *S. aureus* as a model organism for studying the contribution of phage/SaPI transduction to the evolution of plasmid size. Furthermore, additional analysis of the 243 plasmids using CONJscan[20] revealed a strikingly low frequency of conjugative plasmids among the dataset, with only 2% (5/243) of plasmids classified as conjugative (all FATA type), while an additional 5 plasmids were identified as having incomplete conjugation systems (classified as type MOB and type CONJ), suggesting that these could be mobilisable plasmids or are formerly-conjugative plasmids that have lost components.

**Phages limit the size of plasmids for mobilisation by GT**. Three natural plasmids were chosen to investigate whether plasmid size restricts transduction by phages: pC221 is a small-sized (4.6 kb)

plasmid[21], pI258 has an intermediate size (29 kb; NC_013319), while pGO1 is a 54 kb conjugative plasmid[22]. *S. aureus* SH1000 or RN4220 strains lysogenic for the 80α prophage[23] (43.9 kb) and one of the aforementioned plasmids were induced with mitomycin C (MC), activating the prophage, and the transfer of the different plasmids analysed. In support of our hypothesis, no 80α-mediated transduction of the large pGO1 occurred (<1 transducing units [TrU]/ml; Fig. 1), due to its size preventing packaging into phage capsids. Conversely, the intermediate- and small-sized pI258 and pC221 plasmids were transduced by 80α at 533.33 (±188.24) and 70 (±36.06) TrU/ml, respectively (Fig. 1).

**SaPIs modulate the efficiency of plasmid transfer via GT.** To analyse whether SaPIs impact plasmid transfer, we tested the phage 80α-mediated transfer of pC221, pI258 and pGO1 from each of the previous parental lysogenic strains in the presence of different SaPIs. Two different SaPIs, both induced by helper phage 80α, were selected for these experiments: SaPI1 and SaPIbov2[19]. SaPI1 was chosen because the process of remodelling helper phage capsids to produce small SaPI capsids (with DNA packaging capacities of ~15 kb) has been particularly well characterised for this phage-SaPI pairing[16,24]. Following its induction by phage 80α, SaPI1 scavenges phage capsid proteins and utilises two capsid morphogenesis genes (*cpmAB*) to reconstruct capsids with reduced DNA packaging capacity, at around 33% that of their 80α counterpart[24]. Conversely, we selected SaPIbov2 as it is atypically large (27 kb) owing to a transposon insertion in its genome[25], making it too big to fit into small SaPI capsids. SaPIbov2 circumvents this issue by being a natural *cpmB* mutant, abolishing capsid remodelling to small SaPI capsids, and thus exclusively packaging into phage-sized capsids (with DNA packaging capacities of ~45 kb)[26].

The lysogenic strains for phage 80α carrying different combinations of SaPIs and plasmids were MC induced, and the transfer of the different plasmids analysed. Data are presented as log₁₀-transformed absolute TrU per ml of culture, rather than normalised values per $10^9$ PFU, to avoid confounding plasmid transfer rates in SaPI-containing strains where phage titres are reduced because of SaPI interference and where small-sized plasmids may be transferred within SaPI transducing particles. Phage and SaPI titre values for each strain are available in Table S1. Interestingly, although with different efficiencies, the presence of SaPI1 or SaPIbov2 in the donor increased transfer of the small pC221 plasmid compared to the phage alone (Fig. 1a). Similarly, for plasmid pI258 (intermediate size), the presence of SaPIbov2 substantially increased pI258 transfer (Fig. 1b). However, the presence of SaPI1 reduced pI258 transfer by phage 80α (Fig. 1b). Finally, the analysis of conjugative plasmid pGO1 transfer in the presence of the different SaPIs also produced unexpected results. While the SaPI1 lysogen did not transduce pGO1 (Fig. 1c), the 80α SaPIbov2 background gave rise to multiple transductants of pGO1 in the RN4220 recipient (Fig. 1c); a surprising observation given that the size of the plasmid exceeds the phage capsid packaging capacity.

**SaPI TerS enhances the efficiency of plasmid packaging.** Phage or PICI packaging starts with recognition by the phage- or the PICI-encoded small terminases (TerS) of their cognate DNA. Since SaPIs encode their own small terminase subunit (TerS_S), which is functionally analogous but different in sequence to that encoded by the helper phage (TerS_P; Fig. S2), to drive preferential packaging of their own DNA, our previous results suggested that pC221 packaging by the SaPI1-encoded TerS_S is more efficient

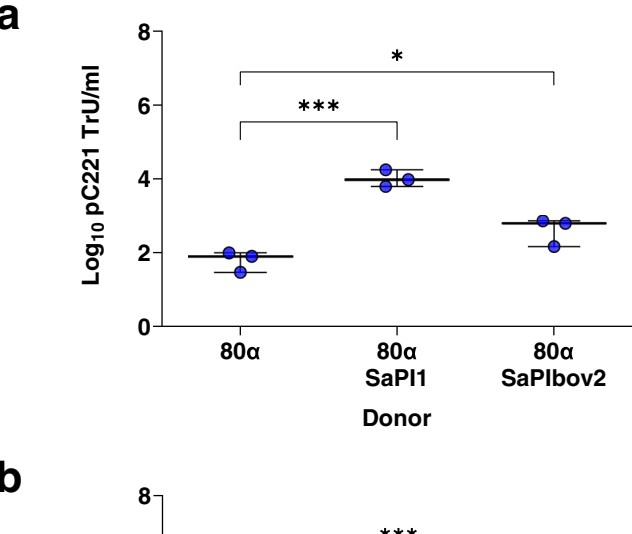

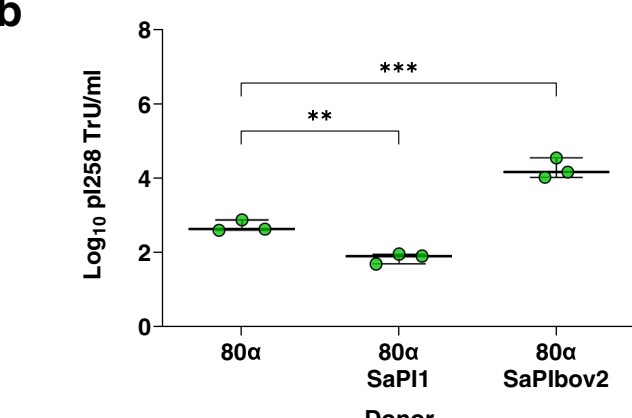

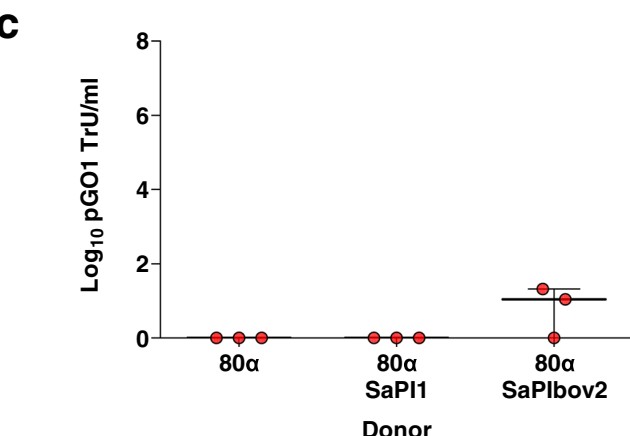

**Fig. 1 Role of phage and SaPIs in plasmid transfer.** *S. aureus* strains lysogenic for phage 80α with or without SaPI1 or SaPIbov2 and carrying plasmids **a** pC221 (4.6 kb; blue circles), **b** pI258 (29 kb; green circles) or **c** pGO1 (54 kb; red circles) were induced with mitomycin C to produce lysates. Log₁₀ transductants (TrU) per ml of lysate were determined for each plasmid in an RN4220 recipient. All data is the result of three independent experiments (*n* = 3). Data for each donor strain are represented as boxplots where the middle line is the median, the lower and upper hinges correspond to the 25th and 75th percentiles, and the whiskers extend from the minimum to maximum values, with all individual data points shown as coloured circles. A one-way ANOVA with Tukey's multiple comparisons test compared mean differences between each strain and the 80α control. Asterisks denote significant adjusted *p* values: **a** \*\*\**p* = 0.0002, \**p* = 0.0355; **b** \*\**p* = 0.0038, \*\*\**p* = 0.0001. All other values were not significant.

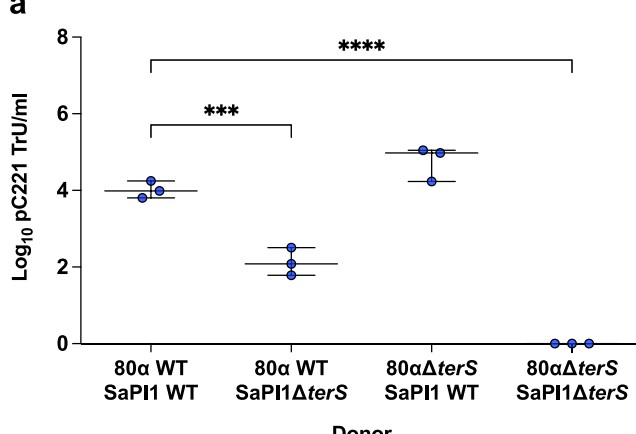

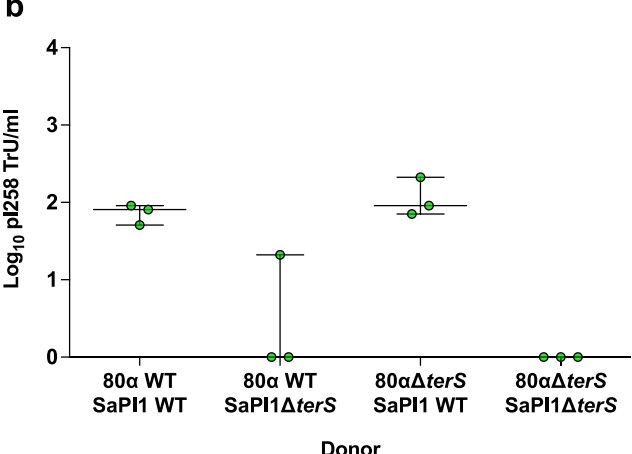

**Fig. 2 SaPI TerS drives efficient packaging of plasmid DNA into transducing particles.** RN4220 strains lysogenic for phage 80α (WT or mutant in small terminase) and SaPI1 (WT or mutant in small terminase), carrying plasmid **a** pC221 (4.6 kb; blue circles) or **b** or pI258 (29 kb; green circles) were induced with mitomycin C to produce lysates. $\log_{10}$ transductants (TrU) per ml of transducing lysate were determined for each plasmid in an RN4220 recipient. All data is the result of three independent experiments ($n = 3$). Data for each donor strain are represented as boxplots where the middle line is the median, the lower and upper hinges correspond to the 25th and 75th percentiles, and the whiskers extend from the minimum to maximum values, with all individual data points shown as coloured circles. For **a**, a one-way ANOVA with Tukey's multiple comparisons test compared mean differences between each strain and the 80α WT SaPI1 WT control. Asterisks denote significant adjusted $p$ values: ***$p = 0.0003$, ****$p < 0.0001$. For **b**, a Kruskal–Wallis (one-way ANOVA on ranks) with Dunn's multiple comparisons test compared mean rank values between each strain and the 80α WT SaPI1 WT control. No statistically significant differences were observed between the groups ($p > 0.05$).

than that of the phage-encoded TerS$_P$. To test this, we repeated the experiments using donor strains carrying mutations in either the 80α *terS* or SaPI1 *terS* genes, or in both. In the presence of the wild type (WT) phage, loss of TerS$_S$ resulted in a 1.9-log reduction in the transfer of plasmid pC221 (Fig. 2a), bringing transfer rates broadly into line with our earlier observations for the ability of the 80α phage to mobilise this plasmid in the absence of a SaPI. Interestingly, transfer of pC221 displayed a 0.7-log increase in the presence of the 80α *terS* mutant phage and WT SaPI1 (Fig. 2a). Together, these results imply that TerS$_S$ is more efficient than phage TerS$_P$ at packaging plasmid pC221 DNA into phage and SaPI capsids. As anticipated, no transfer was observed in the

presence of the double mutant background as capsid packaging is abolished in this strain (Fig. 2a), confirming that the phage and SaPI terminases drive plasmid packaging into capsids for GT.

We also assessed the relative contributions of TerS$_P$ and TerS$_S$ to transfer of pI258 (Fig. 2b). Deletion of SaPI1 TerS$_S$ from the donor strain resulted in a substantial reduction in pI258 transfer compared to the WT control. Conversely, deletion of TerS$_P$ did not produce any marked changes in pI258 transfer compared to the WT donor provided SaPI TerS$_S$ was present. As before, no pI258 transfer was observed from the double *terS* mutant (Fig. 2b). These data suggest that like pC221, pI258 is also packaged more efficiently using the SaPI TerS and that another mechanism must be responsible for the reduction in pI258 transfer observed in the first experiment from the 80α SaPI1 donor background relative to that of the phage alone (Fig. 1b).

To clearly confirm that the SaPI-encoded TerS$_S$ is more efficient than the phage TerS$_P$ in packaging plasmid DNA, we compared the packaging efficiencies of TerS$_P$ and TerS$_S$ for plasmids pC221 and pI258 using inducible plasmids to over-express either the 80α TerS or SaPIbov1 TerS in either the 80α ΔterS pC221 or 80α ΔterS pI258 background. An empty expression plasmid control was also included. Strains carrying the expression plasmids and either pC221 or pI258 were induced at the same time as MC was added to activate the 80α ΔterS prophage, and the resulting lysates were sterile filtered following lysis. pC221 or pI258 transfer from each background was then tested in an RN4220 recipient.

No transfer was observed from the 80α ΔterS pC221 donor background carrying the empty expression plasmid. When TerS$_P$ was overexpressed, pC221 transfer was 4.69 (±0.04) $\log_{10}$ TrU/ml, but when TerS$_S$ was overexpressed, transfer was significantly increased to 5.84 (±0.14) $\log_{10}$ TrU/ml, indicating that SaPIbov1 TerS is more efficient than 80α TerS at packaging plasmid pC221 ($X^2$ test, $p < 0.01$). We used variants of the same expression plasmids with pI258, however we swapped the erythromycin resistance cassette on the inducible plasmids with chloramphenicol resistance in order to enable compatibility with pI258. The results were similar for pI258 to that seen for pC221: no transfer was observed when the empty plasmid was induced, 5.78 (±0.08) $\log_{10}$ TrU/ml were obtained with TerS$_P$ was overexpressed, while 6.29 (±0.09) $\log_{10}$ TrU/ml were obtained when SaPIbov1 TerS was overexpressed ($X^2$ test, $p < 0.05$), indicating that TerS$_S$ was also more efficient than TerS$_P$ at packaging the pI258 plasmid for transduction.

**SaPI capsid modifications restrict plasmid pI258 transfer.** As previously mentioned, we observed decreased pI258 transfer by phage 80α in the presence of SaPI1, which we hypothesised was due to the ability of SaPI1 to strongly promote the formation of SaPI-sized capsids[24], preventing full packaging of pI258. Conversely, because plasmid pC221 is small enough to fit within SaPI-sized capsids, we hypothesised that SaPI1-mediated capsid remodelling would have no effect on its transfer. To test this, we used two complementary strategies: firstly, we utilised a SaPI1 mutant in the *cpmAB* genes, incapable of producing SaPI-sized capsids, and repeated the experiment. Transfer of the pI258 plasmid increased substantially in the presence of the SaPI1 *cpmAB* mutant (Fig. 3b) and was in fact higher than that observed in the presence of SaPIbov2 (Fig. 1b). Surprisingly, pC221 transfer from the SaPI1Δ*cpmAB* donor also appeared to be enhanced (Fig. 3a), however closer inspection of the data indicated that this was an artefact of the fact that this background produced ~1.4-log higher total transducing particles, i.e. phage plus SaPI particles, than the 80α SaPI1 background ($8.92 \times 10^9$ particles/ml for 80α SaPI1Δ*cpmAB*, $3.32 \times 10^8$ particles/ ml for

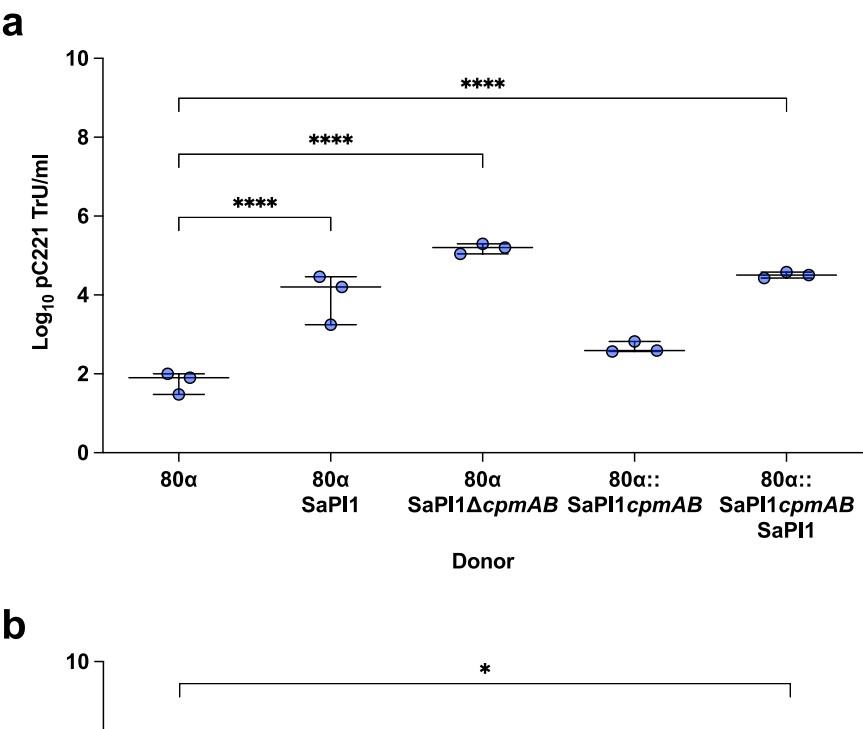

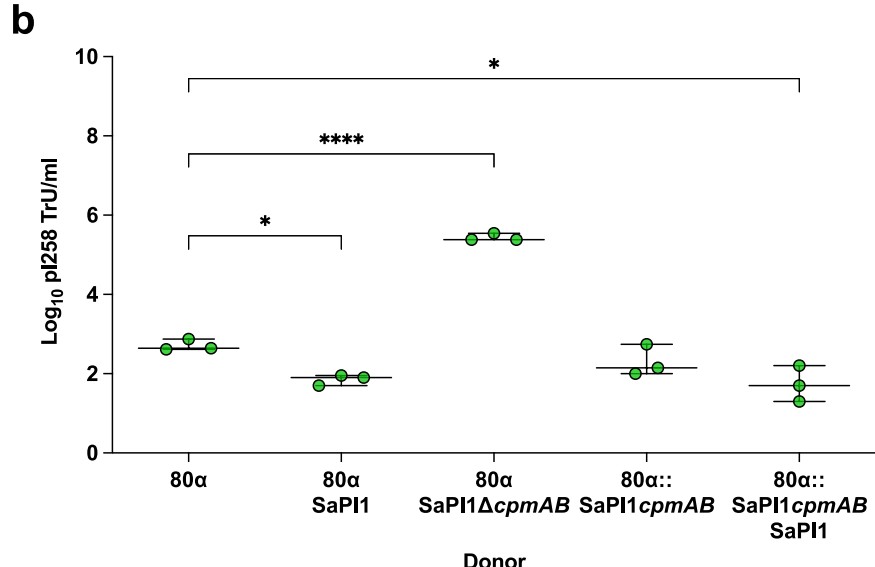

**Fig. 3 Effect of phage capsid remodelling on plasmid transfer.** RN4220 strains lysogenic for phage 80α (WT or carrying the capsid remodelling *cpmAB* genes from SaPI1) with SaPI1 (WT or mutant in *cpm*AB) carrying plasmids **a** pC221 (4.6 kb; blue circles) or **b** pI258 (29 kb; green circles) were induced with mitomycin C to produce lysates. Log₁₀ transductants (TrU) per ml were determined for each plasmid in an RN4220 recipient. SaPI1 mutants in *cpmAB* are incapable of small SaPI-sized capsid production so only produce phage-sized capsids (~45 kb capacity); the 80α::SaPI1*cpmAB* mutant overwhelmingly redirects capsid production to small SaPI-sized particles (~15 kb capacity) incapable of accommodating phage DNA. All data is the result of three independent experiments ($n = 3$). Data for each donor strain are represented as boxplots where the middle line (bold) is the median, the lower and upper hinges correspond to the 25th and 75th percentiles, and the whiskers extend from the minimum to maximum values, with all individual data points shown as coloured circles. A one-way ANOVA with Tukey's multiple comparisons tests compared mean differences between each group and the 80α control. Asterisks denote significant adjusted *p* values: **a** ****$p < 0.0001$; **b** * 80α vs 80α SaPI1 $p = 0.0270$, ****$p < 0.0001$, * 80α vs 80α::SaPI1*cpmAB* SaPI1 $p = 0.0125$. All other values were not statistically significant.

80α SaPI1), which is broadly the difference (1.3 logs) observed for pC221 transduction rates between the two backgrounds, indicating that this difference is likely due to more particles being available in the SaPI1Δ*cpmAB* strain for mis-packaging of plasmid pC221 rather than being an effect of capsid size. Of note is the fact that the production of SaPI-sized capsids is one important mechanism of phage interference, and therefore, in the absence of the SaPI1 *cpmAB* genes phage reproduction increases[24].

In parallel, we analysed transfer of both plasmids after MC induction of the prophage 80α::SaPI1*cpmAB*. In this prophage the

SaPI1 *cpmAB* genes were inserted into the 80α late operon, where they are co-expressed with the structural genes for the 80α virion, promoting a massive production of SaPI-sized particles at the expense of the phage-sized capsids[24]. Note that the data in Fig. 3 are presented as log₁₀-transformed absolute TrU per ml of culture because the number of transducing particles produced by the 80α::SaPI1*cpmAB* background cannot be quantified, preventing normalisation by the total number of transducing particles. In support of the hypothesis, massive production of the SaPI sized particles restricted pI258 but not pC221 transfer (Fig. 3). Additionally, the absence of SaPI1 in the 80α::SaPI1*cpmAB*

donor background reduced pC221 transfer substantially (Fig. 3a), further substantiating our previous results indicating the importance of TerS$_S$ for enhanced pC221 packaging.

**SaPIs select for pGO1 variants with distinct mobility profiles.** Since plasmid pGO1 exceeds phage capsid capacity for packaging, we hypothesised that SaPIbov2 would only be able to transduce remodelled variants of the pGO1 plasmid. To better characterise this process, five evolved plasmids from multiple separate experiments (plasmids pGO1evol$_A$ to pGO1evol$_J$), were selected and sequenced. Genome sequencing revealed that the pGO1evol plasmids have incurred a variety of deletion and rearrangement events that result in a reduction of the overall plasmid size to be compatible with packaging into phage-sized capsids. Figure 4 and Supplementary Data 1 summarise the features of each of these plasmids.

In the cases of plasmids pGO1evol$_A$, pGO1evol$_B$ and pGO1evol$_F$, two separate recombination events occurred, leading to losses of two different regions from each plasmid. For all three plasmids, the first recombination event took place between almost identical (99.4% identity) IS elements located at coordinates 14,227-15,016 (*tnp*A) and 22,925-23,714 (*tnp*C), leading to loss of bleomycin and neomycin resistance derived from pUB110 integration. For pGO1evol$_A$ and pGO1evol$_B$, a second recombination event occurred between identical IS elements at coordinates 45,586–46,375 (*tnp*G) and 53,212–54,000 (*tnp*I), deleting the 7.6 kb pSK639-integrated region. Conversely, pGO1evol$_F$ incurred a second recombination event between identical IS regions 37,328–38,118 (*tnp*D) and 40,892–41,682 (*tnp*E), deleting the 3.5 kb pSK89-integrated region. Interestingly, all three of these plasmids retained the operon encoding the conjugation machinery (*tra*N-O), as well as the *nes* relaxase and *ori*T, permitting them to remain competent for conjugative transfer in addition to being mobilisable by transduction (Table S2).

Surprisingly, examination of pGO1evol$_C$ and pGO1evol$_J$ revealed the formation of two distinct plasmid-SaPI hybrids likely resulting from a recombination event between the IS257 elements present within pGO1 and the transposase present in SaPIbov2. Since SaPI packaging of plasmids is size-limited, we speculate that these hybrids were formed by a reduction in pGO1 size due to recombination between the IS257 elements present on the plasmid, with a second recombination event between the plasmid and SaPI ISs leading to integration of the plasmid sequence into the SaPI on the bacterial chromosome (Fig. 4c). For pGO1evol$_C$, the initial recombination event appears to have occurred between IS257 elements at coordinates 53,212–54,000 (*tnp*I) and 37,328–38,118 (*tnp*D), deleting more than 38 kb from plasmid pGO1. A subsequent recombination event between the 96% identical IS sequences on pGO1evol$_C$ (*tnp*D) and the SaPI (*tnp*SBov2) has facilitated integration of the remaining 15 kb fragment of pGO1 into SaPIbov2, generating a hybrid 40.2 kb island encoding trimethoprim and quaternary ammonium compound resistances, as well as mobilisation-associated factors. Conversely, pGO1evol$_J$ has been formed by two distinct deletion events: firstly, recombination between IS257 elements at coordinates 22,925–23,714 (*tnp*C) and 45,586–46,375 (*tnp*G) of pGO1, resulting in the deletion of more than 31 kb from the parental plasmid; secondly, loss of the pSK89-integrated region flanked by IS elements at coordinates 37,328–38,118 (*tnp*D) and 40,892–41,682 (*tnp*E). The resulting 19 kb sequence has then been integrated into the SaPIbov2 sequence on the bacterial chromosome through recombination between 97.6% identical IS regions on the derivative plasmid (*tnp*C) and SaPIbov2 (*tnp*SBov2), creating a 43 kb plasmid-SaPI hybrid featuring gentamicin resistance and encoding the entire

conjugative transfer gene operon, though without the *nes* relaxase or *ori*T. Transfer of these hybrid elements was extremely efficient by phage 80α transduction, but neither could be mobilised by conjugation (Table S2).

Our previous results suggested that SaPIbov2 promotes the transduction of pGO1-derivative plasmids because it does not produce small SaPI capsids and because the SaPI TerS$_S$ is more efficient than the phage TerS$_P$ at packaging pGO1 (as previously demonstrated for the other plasmids). To test this, pGO1 transfer was tested in the 80α SaPI1Δ*cpm*AB mutant, which encodes TerSs but does not produce small SaPI-sized capsids. Three transductants (designated pGO1evol$_E$, $_H$ and $_I$) across multiple independent experiments were obtained. Sequencing of pGO1evol$_{E-I}$ (Fig. 4, Supplementary Data 1) revealed that these plasmids had evolved via recombination of two IS257 elements located at coordinates 14,227-15,016 (*tnp*A) and 37,328-38,118 (*tnp*D), leading to loss of bleomycin and neomycin resistance, as well as the conjugative transfer operon; though retaining the plasmid relaxase (*nes*) and *ori*T region.

Interestingly, despite substantially lower efficiency than for that of the SaPI-containing donor strains, three identical evolved plasmids (pGO1evol, pGO1evol$_2$ and pGO1evol$_3$) were eventually obtained from independent experiments following many attempts to transduce pGO1 from the SH1000 80α donor. Genome sequencing identified pGO1evol as a 27,574 bp plasmid corresponding to bases 41,125–14,698 of plasmid pGO1, with recombination between two 96% identical IS257 elements at coordinates 40,892–41,682 (*tnp*E) and 14,227–15,016 (*tnp*A) of pGO1 (Fig. 4a). pGO1evol$_2$ and pGO1evol$_3$ also exhibited reduction via recombination between these sites. These data are consistent with others' observations that IS elements facilitate distinct remodelling strategies for the evolution of medium-to-large composite plasmids[27].

**SaPI-driven remodelling of pGO1 exchanges plasmid mobility.** Sequence analyses of the pGO1-evolved plasmids suggest that some of them have lost their conjugation cluster, suggesting their mobility is expected to be absolutely dependent on phage transduction. To confirm loss of pGO1evol conjugation, while demonstrating that this loss of mobility can be rescued by transduction, competitive conjugation and transduction assays were performed. Following competitive mating between pGO1 and the conjugation-defective plasmids pGO1evol$_E$, pGO1evol$_I$ and pGO1evol, all transconjugants displayed dual resistance to gentamicin and neomycin, indicating carriage of pGO1, confirming loss of conjugative transfer ability in the evolved plasmids. Curiously, despite displaying a higher conjugative transfer efficiency alone than that for pGO1 (Table S2), pGO1evol$_F$ transfer was less efficient than pGO1 when the plasmids were placed in direct competition, with all transconjugants displaying the antibiotic profile consistent with pGO1 (Table S3), suggesting that pGO1evol$_F$ has incurred a cost associated with its size reduction relative to that of its parent plasmid that is only apparent under specific conditions. Conversely, when placed in competition with pGO1, pGO1evol$_A$ transferred more efficiently than its parental plasmid, with the evolved plasmid comprising almost 80% of the transconjugant population (Table S3).

Following their isolation, all pGO1-derivative plasmids and SaPI-hybrids were confirmed to be compatible with transduction by phage 80α, with the pGO1-SaPIbov2 hybrids transducing into recipient RN4220 cells at particularly high frequencies (Table S2). In order to address the competitive advantage of pGO1evol of having severely reduced its size compared to the parent plasmid, competitive transduction assays were performed by MC-inducing a mixed culture of *S. aureus* SH1000 80α lysogens harbouring

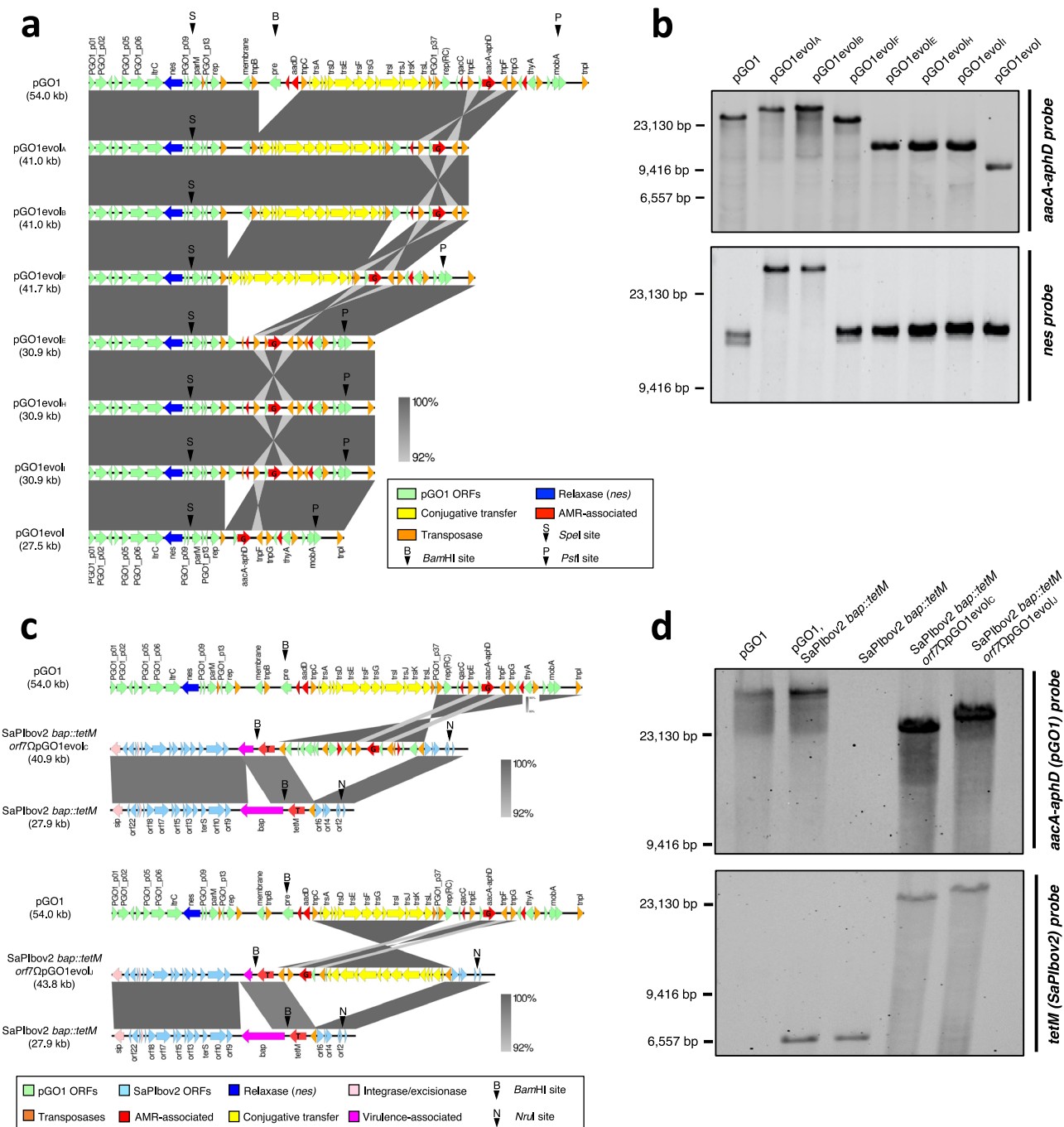

**Fig. 4 Genetic maps of evolved pGO1 and pGO1-SaPIbov2 hybrid species. a** Sequence alignments of circular pGO1-derivative plasmids were performed using Easyfig software (v2.2.2). Genes are coloured according to their sequence and function. Black arrows indicate relevant restriction sites. **b** Plasmid sequences were affirmed using Southern blotting of total DNA digested with *Spe*I, *Bam*HI and *Pst*I, with probes against the *aac*A-*aph*D ['G'] and *nes* regions encoding gentamicin resistance and the relaxase, respectively. Expected plasmid fragment sizes following triple digestion were as follows: **pGO1**, G: 30,498 bp, *nes*: 14,103 bp; **pGO1evol~A-B~**, G: 41,026 bp, *nes*: 41,026 bp; **pGO1evol~F~**, G: 27,634 bp, *nes*: 14,103 bp; **pGO1evol~E, H-I~**, G: 16,796 bp, *nes*: 14,103 bp; and **pGO1evol**, G: 13,471 bp, *nes*: 14,103 bp. **c** Sequence alignments of SaPIbov2 *bap*::*tet*M *orf7*ΩpGO1evol~C~ and SaPIbov2 *bap*::*tet*M *orf7*ΩpGO1evol~J~ hybrid species were performed using Easyfig software (v2.2.2). Genes are coloured according to their sequence and function. Black arrows indicate relevant restriction sites. **d** SaPIbov2 *bap*::*tet*M orf7ΩpGO1evol sequences were affirmed using Southern blotting of total DNA digested with *Bam*HI and *Nru*I, with probes against the *aac*A-*aph*D region encoding gentamicin resistance [G] in pGO1, and the *tet*M cassette in SaPIbov2 *bap*::*tet*M [T], respectively. Expected fragment sizes following double digestion were as follows: **pGO1**, G: 54,000 bp, T: no band; **SaPIbov2 bap::tetM**, G: no band, T: 6,488 bp; **SaPIbov2 bap::tetM orf7ΩpGO1evol~C~**, G and T both 22,317 bp; and **SaPIbov2 bap::tetM orf7ΩpGO1evol~J~**, G and T both 25,584 bp. See Supplementary Data 1 for details of recombination events and Table S2 for details of mobility mechanisms utilised by the variant pGO1 plasmids. See Supplementary Data 1 for details of recombination events, Table S2 for details of mobility mechanisms utilised by the variant pGO1 plasmids, and Table S3 for details of relative fitness of the variant pGO1 plasmids compared to the parental pGO1. Panels **b** and **d** are representative of data obtained from three independent experiments ($n = 3$) yielding similar results.

either pGO1 or pGO1evol to produce a lysate, which was used to transduce an RN4220 recipient. A total of 110 transductants were obtained over multiple replicate plates from three independent experiments. Neomycin resistance was assessed for each of the 110 transductants and all showed sensitivity to neomycin, indicating pGO1evol transduction only. This was further confirmed by PCR using primers specific within the shared region of pGO1 and pGO1evol, and primers that amplify across the deleted region of pGO1evol. The results are in line with the hypothesis of this study, in that larger conjugative plasmids, when reduced in size, can retain mobility due to rescue by phage transduction.

**Phages and PICIs permit plasmid transfer between species**. Previous studies have demonstrated that conjugation plays an important role in the intra- and inter-species transfer of plasmids between bacterial species, promoting the emergence of novel multi-resistant bacterial clones[28]. However, whether PICIs and phages can promote intra- or intergeneric plasmid transfer in nature remains undetermined. Since transducing phages can transfer SaPIs to non-*aureus* staphylococci and to the Gram-positive pathogen *Listeria monocytogenes*[26,29,30], we reasoned that transducing phages and PICIs might also be capable of transferring plasmids intra- and intergenerically. We tested this with plasmids pC221 and pI258.

To assess the capacity for interspecific and intergeneric plasmid transfer following antibiotic-mediated prophage induction, these strains were exposed to sub-inhibitory concentrations of the antibiotic ciprofloxacin, activating the SOS-response, and the resulting lysates tested for plasmid transfer. To test for trans-specific or trans-generic transduction, coagulase-negative staphylococci species and *L. monocytogenes* strains were used as recipients for plasmid transfer. Plasmid pC221 was transferred to *S. xylosus* and *L. monocytogenes* strains at frequencies only slightly lower than to *S. aureus* (Fig. 5a). pC221 transfer to *S. epidermidis* was also observed, though at a much lower frequency than for the other recipients, and only in the presence of SaPI1. Interspecific transfer of pI258, occurring at the same rate as that for *S. aureus*, was observed in *S. xylosus* (Fig. 5b). Restriction digestion analysis confirmed that the complete plasmids were transferred to the recipient strains. In contrast, deletion of the SaPI- and phage-encoded TerS eliminated transfer (Fig. 5). Together, these results show that antibiotic-mediated SOS-induction of prophages permits intra- and intergeneric transfer of the plasmids via phage and PICIs.

**Phage- and PICI-mediated plasmid transfer occurs in milk**. Bulk milk tanks typically contain an array of different bacterial species that inhabit the dairy niche[31]. Importantly, milk tanks may also contain traces of antibiotics, present at subinhibitory concentrations[32], which promote prophage and PICI induction[33]. Accordingly, milk represents a more natural model environment for examining phage- or PICI-mediated plasmid transfer. To test this, we used as donor either *S. aureus* JP21924, which is a non-lysogenic strain that carries SaPI1 *tst*::*tet*M and plasmid pC221, or JP21925, a derivative strain of JP21924 lysogenic for phage 80α. Note that donor strains are tetracycline (SaPI-encoded) and chloramphenicol (pC221-encoded) resistant. The rifampicin-resistant *S. aureus* strain JP21894 was used as recipient. Donor and recipient strains were individually incubated in milk containing subinhibitory concentrations of the antibiotic cipro-floxacin to activate the prophages[34]. Next, equal amounts of the donor and recipient strains were mixed and then plated onto TSA containing rifampicin (recipient resistance) and chloramphenicol (plasmid resistance). Chloramphenicol-resistant recipient strains

were obtained only when the lysogenic strain JP21925 was used as donor (Fig. 6). Importantly, none of the resultant colonies were tetracycline resistant, confirming that the plasmid moved from the donor to the recipient strain.

**Small plasmids are enriched in SaPI-containing genomes**. Since our previous results indicated that the phage- and SaPI-mediated transfer of the mobilisable and non-transferable plasmids occur in nature, we hypothesised that it would be possible to obtain additional evidence of the relevance of these processes in vivo by interrogating the genome databases. Transduction is primarily mediated by temperate phages carrying the *pac* site–headful system for DNA packaging[18]. Temperate phages have a lysogenic lifecycle where they remain as prophages integrated in the bacterial genome (collectively termed lysogens) until induction initiates a lytic lifecycle with phage particles being produced. Bacterial cells that can resist lytic phage activity are likely to be transduced more efficiently than those susceptible to phage-mediated killing[35]. Since prophages protect their host cell from infecting phages, so-called lysogenic immunity, transduction primarily occurs in cells that have been previously lysogenised[36]. Thus, we hypothesised that: i) the frequency of genomes carrying plasmids should increase in lysogenic strains; and ii) the presence of small plasmids (<15 kb) should be increased in *S. aureus* strains carrying SaPIs.

To test these hypotheses, we analysed the distribution of phages, SaPIs, and plasmids across 295 *S. aureus* genomes. Since most genomes contained at least one prophage (284/295), we couldn't test the first hypothesis in this bacterial species. Interestingly, 61% (180/295) of genomes contained at least one SaPI, demonstrating the widespread nature of these elements in *S. aureus*. Remarkably, analysis of the genomes revealed a significant enrichment of genomes carrying both SaPIs and plasmids smaller than 15 kb (Yates corrected $X^2 = 5.153$, df = 1, $p = 0.0232$) (Fig. 7), with 74% (50/68) of small plasmid-carrying genomes carrying also a SaPI element. These results suggest that small plasmids may be enriched in SaPI-carrying genomes because their survival is favoured in SaPI-rich communities that permit their horizontal transfer.

**Poly-lysogenic strains have higher numbers of plasmids**. To investigate whether the frequency of genomes carrying plasmids should increase in lysogenic strains, we analysed the distribution of plasmid/prophage in 8903 bacterial genomes from NCBI. Plasmids ranked according to size displayed a bimodal distribution, with the larger peak correlating with the peak in prophage genome size (Fig. 8a–c), suggesting that most plasmids are appropriately sized to be compatible with packaging into transducing phage capsids. We also observed that: (i) the frequency of genomes carrying plasmids increases with the number of phages per genome (Fig. 8d); and (ii) the presence of prophages does not increase in strains carrying multiple plasmids (Fig. 8e). Additionally, our analysis suggested that there are fewer genomes carrying a plasmid but not a prophage than would be expected due to chance (observed = 262, expected = 422.4; observed/expected = 0.62, X-squared = 63.971, df = 1, $p = 1.263e^{-15}$). These results could suggest that phages play a role in dictating the distribution of plasmids in a bacterial population, but that plasmids do not reciprocally dictate the distribution of phages, which would be consistent with the hypothesis that phages facilitate mobilisation of plasmids in bacterial populations. However, other factors apart from phage-mediated mobilisation, such as for example differences in host defence systems, may help to explain the co-occurrence of phages and plasmids in genomes, and further work will be required to confirm our hypothesis.

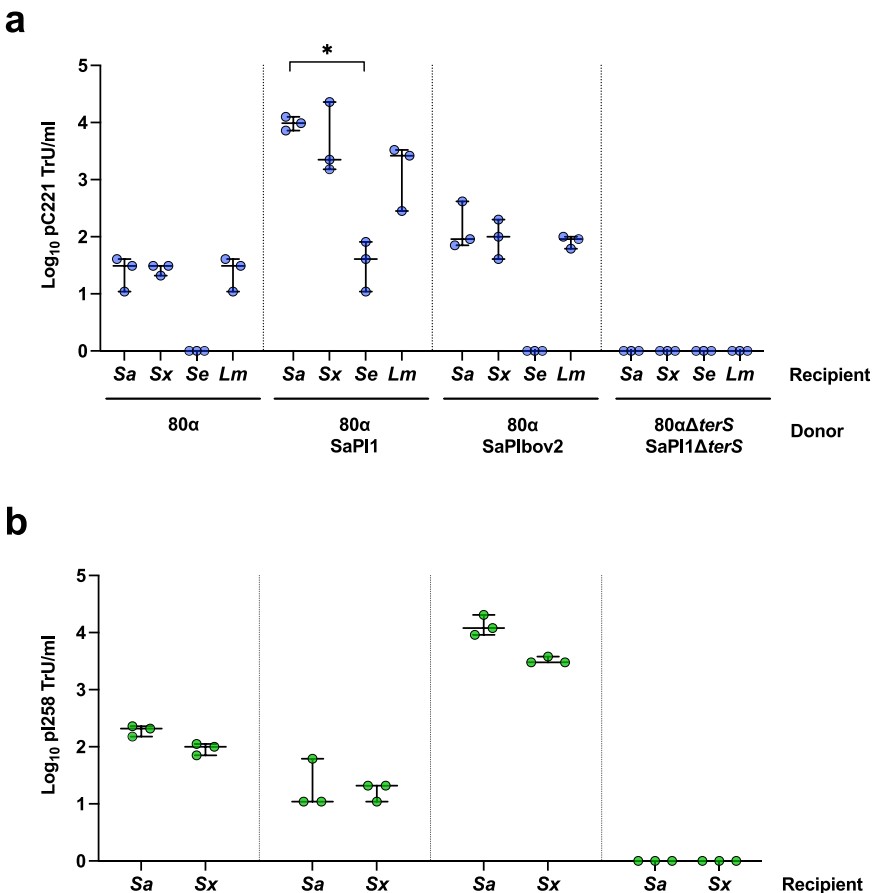

**Fig. 5 Intra- and intergeneric transfer of pC221 and pI258.** RN4220 strains lysogenic for phage 80α (WT or mutant in small terminase) with SaPI1 (WT or mutant in small terminase) or SaPIbov2 (WT), carrying plasmids **a** pC221 (4.6 kb; blue circles) or **b** pI258 (29 kb; green circles) were induced with ciprofloxacin at 0.6 μg/ml to produce lysates. $Log_{10}$ transductants (TrU) per ml were determined for each plasmid in different recipient strains: *S. aureus* RN4220 (*Sa*), *S. xylosus* (*Sx*), *S. epidermidis* (*Se*), and *L. monocytogenes* (*Lm*) for plasmid pC221, and *S. aureus* RN4220 (*Sa*), and *S. xylosus* (*Sx*) for pI258 (as this plasmid does not replicate in *S. epidermidis* or *L. monocytogenes*). All data is the result of three independent experiments (*n* = 3). Data for each donor strain are represented as boxplots where the middle line (bold) is the median, the lower and upper hinges correspond to the 25th and 75th percentiles, and the whiskers extend from the minimum to maximum values, with all individual data points shown as coloured circles. For **a**, Kruskal–Wallis (one-way ANOVA on ranks) with Dunn's multiple comparison tests compared mean rank values between *S. aureus* and each recipient strain within each donor group. Asterisks denote significant *p* values: **a** \**p* = 0.0001; all other values were not statistically significant. For **b** Mann–Whitney tests (two-tailed) compared rank values between *S. aureus* and *S. xylosus* within each donor group. No statistically significant differences were observed (*p* > 0.05).

## Discussion

Plasmid persistence in bacterial populations is maintained as a finely balanced equilibrium between vertical and horizontal modes of transmission. The ability of a plasmid to pay its metabolic 'rent', so to speak, demands that a balance be achieved between being large enough to carry many potentially beneficial accessory genes, whilst also not encumbering the cell with an unsustainable metabolic burden. Horizontal transfer of plasmids by conjugation, either autonomously or facilitated in trans, is often considered to be the major mechanism of plasmid transmission, yet horizontally-acquired plasmids have been shown to impose a fitness cost upon their host cell[37–41], indicating that the benefits of horizontal transfer for these plasmids may be somewhat mitigated by a cost to their vertical transmission, though the accessory genes carried by these large plasmids may determine that this cost is still of net-benefit to the host cell under certain conditions. Indeed, Slater et al. found that it is beneficial for plasmids to be larger in complex environments, allowing for carriage of more genes for adaptive traits[42]. It is, therefore, perhaps surprising that plasmids encoding the transfer machinery

required for plasmid DNA translocation are under-represented among the sequenced *S. aureus* plasmid genomes[43,44]. This relative scarcity of conjugative plasmids capable of facilitating in trans transfer of mobilisable plasmids raised the question of how these mobilisable plasmids, encoding either MobA or Pre, or *ori*T sequences, are transferred in nature. The same question applies to the non-transmissible plasmids, which are well-represented in bacterial genomes, with Ramsey et al.[45] revealing that one-fifth (52/259, 20.1%) of *S. aureus* plasmids in the phage-transducible range (<45 kb) carry neither a replicative relaxase (for autonomous transfer), nor an *ori*T, MobA or Pre homologue for mobilisation in trans, while almost one-third (40/129, 31.0%) of plasmids within the SaPI-transducible range (<15 kb) harbour neither a relaxase, MobA or Pre homologue, nor *ori*T.

While it is well-established that phages can transduce plasmids under laboratory conditions, the real implications of this have remained poorly understood until now. Here we have used a combination of experimental models and bioinformatic analysis to determine how mobilisable and non-transmissible plasmids move and demonstrate that phages and PICIs are potentially key

players in promoting both intra- and inter-species plasmid transfers in nature. Since in *S. aureus* and in other species plasmids encode antibiotic resistance genes, our results reveal additional routes involved in the emergence of multi-resistant

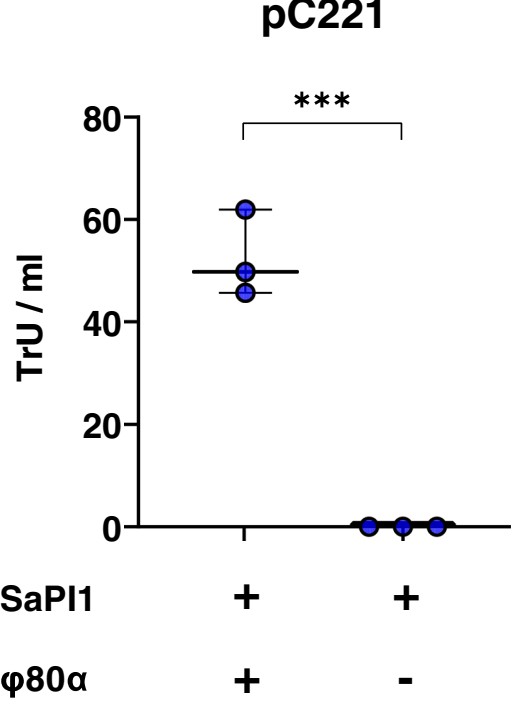

**pC221**

**Fig. 6 Phage- and PICI-mediated plasmid transfer occurs in milk.** Donor strains (RN4220 SaPI1+ lysogenic or not for phage 80α) and recipient strain (RN4220 spontaneous rifampicin-resistant mutant) were incubated independently at 32 °C in milk containing ciprofloxacin. After 6 h, each donor strain was mixed with the recipient strain and plated on selective agar media. The figure shows the number of transductants per ml of donor strain detected. All data is the result of three independent experiments. No transductants were detected using the non-lysogenic donor strain. Plots indicate the median (bold bar) and range for each donor strain with individual data points shown as coloured circles. ***$p = 0.0004$, *t*-test (two-tailed). $n = 3$ experimental replicates.

bacterial clones. Importantly, our results also change the way transduction and conjugation have been seen in the past as the main drivers of bacterial evolution. Transduction and conjugation are often considered as alternative strategies for the horizontal transfer of plasmids. Our results demonstrate that in the case of the conjugative plasmids, whose size usually exceeds the limit to be packaged within a phage capsid, this perception must be considered correct. However, the study of the mobilisable plasmids challenges this dichotomy by demonstrating that transduction and conjugation are inherently coupled processes that are shaped by a dynamic interplay between phage, plasmids, and PICIs. Our results demonstrate that in *S. aureus*, mobilisable plasmids can significantly increase their transferability by exploiting phages, PICIs and conjugative plasmid machineries for transfer, highlighting the fascinating strategies open to these plasmids to permit spread in nature.

Our results also demonstrate that interspecies and intergeneric mobilisation of plasmids is not exclusively mediated by conjugation but may also occur via transduction. As bacterial pathogens become increasingly antibiotic resistant, lytic phages have been proposed for phage therapy on the grounds that they would not introduce adventitious host DNA into target organisms and that the phages are so restricted in host range that the resulting progeny are harmless and will not result in dysbiosis of human bacterial flora. Because plaque formation was once thought to determine the host range of a phage, the evolutionary impact of phages on the bacterial strains they can transduce, but are unable to parasitize, has remained an unrecognised aspect of phage biology and pathogen evolution. Our results add to the recently recognised concept of "silent transfer" of pathogenicity and antibiotic resistance factors carried by MGEs[26,29,30,46] by phages that cannot grow on the target organism. It should be noted, however, that while phylogenetic analysis has indicated that intra- and intergeneric phage-mediated transfer events do occur in natural bacterial populations, they do so at extremely low frequencies[47], suggesting that while the potential for phage and/or PICIs to contribute to plasmid transfer within polymicrobial communities exists, it is unlikely to occur at significant rates.

Interestingly, the results presented here account for the observations of others where ARGs have been detected during sampling in virome studies[48–51]. Colombo et al. showed that the distribution of ARGs had strikingly similar profiles among microbial and viral sample fractions[49], strongly implicating a relationship between the two populations, suggesting that

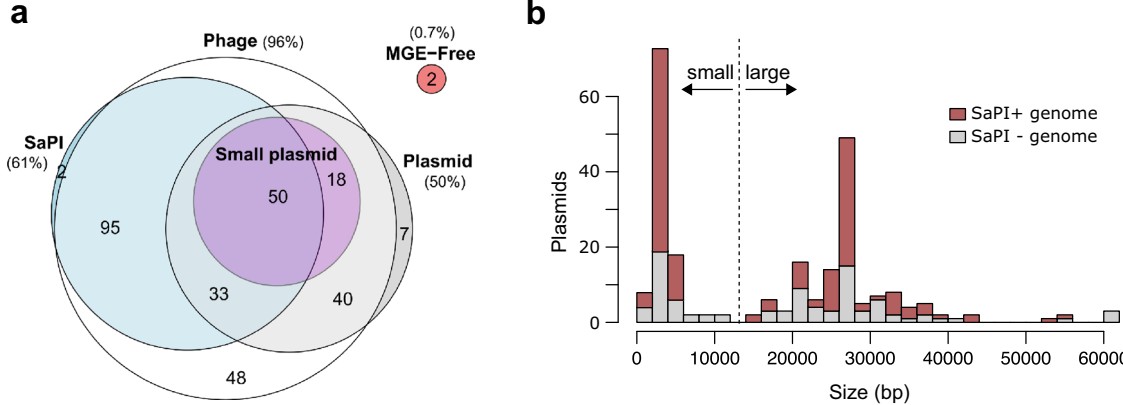

**Fig. 7 Small plasmids are enriched in SaPI-containing genomes. a** Venn diagram indicating *S. aureus* genomes carrying different MGE ($n = 295$ genomes). The frequency of genomes carrying each particular MGE type is denoted by (%), while the numbers indicate the absolute number of genomes that fall into each category. See also Supplementary Data 2. **b** Distribution of sizes of the plasmids present in the *S. aureus* genomes analysed in this study. The colour code indicates whether the plasmids were present in genomes which also contained SaPIs (red) or not (grey). The vertical dashed line indicates the size limit between "small" and "large" plasmids.

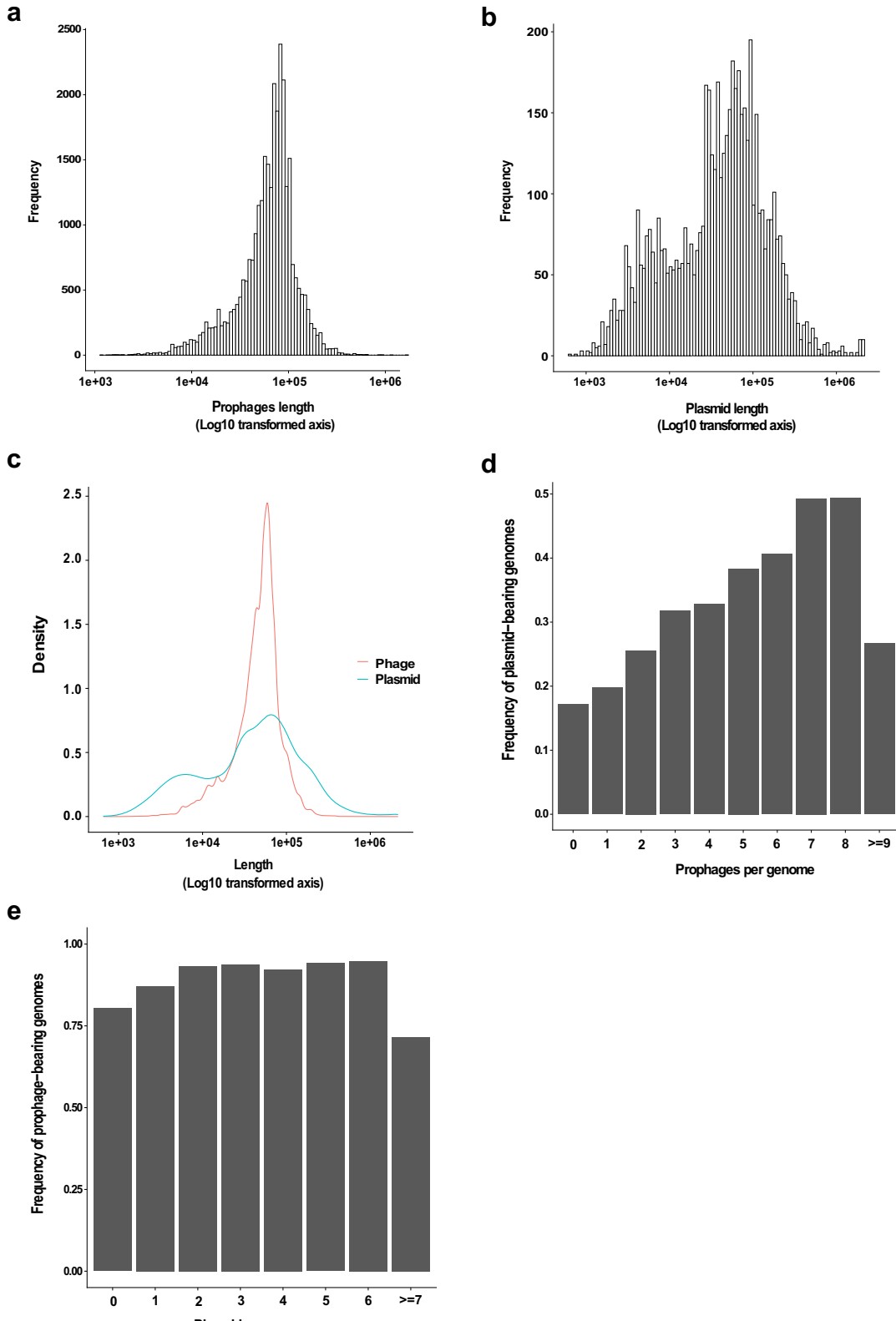

**Fig. 8 Relationship between prophage and plasmid carriage in bacterial populations.** 8903 complete bacterial genomes from the NCBI database were analysed for the presence of plasmids and prophages. **a** Size distribution analysis of prophages ranked according to length. **b** Size distribution analysis of plasmids according to length. **c** Comparison of the length distribution of prophages and plasmids. **d** The frequency of plasmid-bearing genomes was ranked according to the number of prophages per genome, with the maximum rank of 9 being equal to or greater than 9 prophage genomes per record. Slope: $y = 0.027$, indicates a positive correlation between increasing plasmid frequency with increasing prophages. **e** The frequency of prophage-bearing genomes was ranked according to the number of plasmids per genome, with the maximum rank of 7 being equal to or greater than 7 plasmids genomes per record. Slope: $y = -0.0026$, indicates no clear relationship between increasing prophage frequency and increasing plasmid carriage.

detection of these genes in the virome could be attributed to plasmid DNA packaged within phage or PICI capsids. Thus, transducing phages can have an important role in spreading MGEs carrying virulence and resistance genes. We predict here that other MGEs will take advantage of either the PICI- or the phage-mediated transduction as a strategy to promote their spread, both intra- and inter-generically.

We have previously demonstrated that the TerS proteins encoded by either the helper phage or the SaPIs have distinct specificities for sequences contained within highly transferred regions of the bacterial chromosome, indicating that SaPI-mediated transduction occurs as a complementary process to phage-mediated GT[52]. Here, we have not just demonstrated this effect in a naturally occurring system, but our results have revealed an enhanced affinity of SaPI TerS proteins for packaging *S. aureus* plasmids. Remarkably, enhanced plasmid packaging driven by SaPI TerS serves as an example of opportunistic molecular parasitism of the SaPI by the plasmid, thus essentially demonstrating parasitism of a parasite. Importantly, however, we observed that the advantage of TerS-enhanced packaging for plasmids >15 kb was mitigated somewhat by the effect of capsid remodelling interference, driven in the case of SaPI1 by the *cpm*AB genes, which could be an underlying reason for the enrichment of small plasmids (<15 kb) in SaPI-containing genomes revealed by our bioinformatic analysis. More generically, since most of the PICIs described to-date have the ability to promote the formation of PICI-sized capsids[11,12,53], it is interesting to speculate that PICIs may also drive the evolution and transfer of small-sized plasmids in species other than *S. aureus*. Importantly, since plasmids are transduced to strains carrying either prophages and/or PICIs, it is also clear that these elements will also benefit from the cargo encoded in the transduced plasmids[36].

Our results have also demonstrated that phages and SaPIs not just promote plasmid transfer, but may also contribute to plasmid evolution in *S. aureus*, by selecting for variant plasmids that have undergone IS-mediated remodelling, as has previously been described for mosaic-like plasmids in *S. aureus*[27], to sizes compatible with packaging into either the phage or SaPI-sized capsid, with implications for plasmid gene content due to reduced genome size. Notably, although remodelled plasmids are more able to be transferred to new hosts by SaPIs, it is important to emphasize that the ability of these plasmids to successfully spread and persist in their new hosts will depend on the fitness costs and benefits associated with these reduced elements. Interestingly, plasmid remodelling typically occurred through the loss of IS elements carrying accessory genes (in this case, resistance genes) that are likely to provide both fitness costs and benefits. While the absence of these accessory genes likely reduces the prospect that these novel elements spread to high frequency in their new hosts, the loss of costly accessory genes should make it easier for these elements to persist as cryptic parasites in their new hosts populations over the long term.

Finally, we have also shown that IS elements can facilitate recombination between plasmids and SaPIs, generating hybrid MGEs with extremely high transduction efficiencies. SaPIbov2 features a transposon carrying the biofilm-associated protein (*bap*) gene. Fascinatingly, in some species, *bap* has been shown to be plasmid-encoded: plasmid pACK2 from *Staphylococcus simulans* encodes *bap* and its associated transposase[54], leading us to speculate that this gene has been acquired by SaPIbov2 from a co-resident plasmid in its ancestral host. Furthermore, the position of the SaPIbov2 transposon downstream of the regulatory, interference and packaging genes, facilitates integration of plasmid-encoded factors via IS257 recombination at the 3' end of the SaPI, where most accessory virulence genes are located in

SaPIs observed in nature[55], indicating that plasmids represent an important reservoir for acquisition of virulence and antibiotic resistance genes by highly transducible PICI elements. Additionally, we observed that plasmid pGO1evol_J integrated into SaPIbov2 in such a way as to introduce the conjugative transfer (*tra*) genes from pGO1 into the island, forming a structure strikingly reminiscent of the Integrated Conjugative Element (ICE) family of MGEs[56], suggesting that spontaneous recombination between PICIs and plasmids may account for the formation of ICEs in nature.

Overall, we propose that in highly lysogenic species such as *S. aureus*, horizontal transfer via transduction is favoured for plasmids where there is a high benefit to the plasmid in being small. In some circumstances, however, the benefit conferred to the host cell by the plasmid may be reduced owing to fewer genes being carried, thus potentially reducing plasmid fitness in terms of vertical transmission. Intracellular competition for limited capsids between co-habiting mobile elements creates the potential for antagonistic co-evolution between plasmids, phage and SaPIs. For example, the increased efficiency of plasmid packaging by the SaPI terminase suggests that plasmids have evolved to maximise packaging at the expense of SaPIs. This plasmid adaptation is intuitive, as exploiting SaPIs and phage increases the efficiency of plasmid transfer during stress. Furthermore, the low copy number of plasmids relative to the extremely high number of available phage and SaPI capsids means that plasmid hijacking of phage or SaPI capsids has negligible impact on the reproductive success of phage and SaPIs, implying that SaPIs and phage are under very weak selection to evolve counter-adaptations that prevent plasmid packaging. Here, we have unveiled a fascinating example of multi-faceted parasitism: SaPIs parasitise phage particles for their own transduction by utilising island-encoded TerS proteins to drive SaPI DNA packaging into modified phage capsids, to the detriment of the phage; then, in turn, these TerS_S proteins are exploited by natural plasmids to mediate their own dissemination in *S. aureus*. This complex interplay between elements ultimately culminates in a potentially important role for phages and SaPIs in dictating the distribution of plasmids in *S. aureus* on account of mediating their transfer by generalised transduction.

## Methods

**Bacterial strains and growth conditions.** The bacterial strains and plasmids used in this study are detailed in Supplementary Table 4. *S. aureus* was grown in Tryptic soy broth (TSB) or on Tryptic soy agar (TSA) plates supplemented with 1.7 mM sodium citrate. Where appropriate, the following antibiotic concentrations were used for plasmid, SaPI, or chromosomal marker selection: gentamicin, 10 μg/ml; cadmium chloride (CdCl$_2$), 100 μM; chloramphenicol, 10 μg/ml; erythromycin, 10 μg/ml; rifampicin, 10 μg/ml; tetracycline, 3 μg/ml; trimethoprim, 20 μg/ml.

**Phage lysate generation and plasmid transduction titrations.** From overnight cultures, *S. aureus* strains were sub-cultured into TSB, plus appropriate selective antibiotics, and grown at 37 °C, 120 rpm to OD$_{540}$ 0.24 ± 0.02. Prophages were induced with 2 μg/ml mitomycin C and incubated at 30 °C, 80 rpm until completely lysed, before filtering with a 0.2 μm syringe filter (Sartorius). For transductions, 100 μl of phage lysate was used to infect 1 ml of recipient RN4220 supplemented with 4.4 mM CaCl$_2$. After static incubation for 20 min at 37 °C, 3 ml TTA (TSA top agar; 3 g/100 ml TSB, Oxoid, plus 0.75 g/100 ml agar, Formedium) was added to each transduction reaction and the entire contents were poured onto a TSA plate supplemented with 1.7 mM sodium citrate + appropriate selective antibiotics. Plates were incubated for 18-20 h at 37 °C and the colonies were enumerated.

**Phage titrations.** For phage titration, RN4220 recipient cells were grown to exponential phase and adjusted to 0.35 OD$_{540}$. These were mixed with 3.5 ml phage top agar (PTA; 2 g/100 ml Nutrient Broth No. 2, Oxoid, plus 0.35 g/100 ml agar, Formedium, and 10 mM CaCl$_2$), and overlaid onto phage base plates (2 g/100 ml Nutrient Broth No. 2, Oxoid, plus 0.7 g/100 ml agar, Formedium, and 10 mM CaCl$_2$). 10-fold serial dilutions (no dilution to 1 × 10$^7$) of 80α phage lysates were made in phage buffer (100 mM NaCl, 50 mM Tris (pH8), 1 mM MgSO$_4$, 4 mM CaCl$_2$) and triplicate ten microliter spots were dropped onto the recipient lawn.

Plates were incubated at 37 °C for 18-20 h to enable enumeration of phage plaque-forming units (PFU) per ml of lysate.

**WGS analysis of pGO1-derivative plasmids.** Putative pGO1-derivative plasmid sequences were obtained via WGS of the host bacterial strains. Genome sequencing was provided by MicrobesNG (http://www.microbesng.uk) which is supported by the BBSRC (grant number BB/L024209/1). Plasmid-containing strains were prepared for sequencing in exact accordance with the company's instructions. Sequence assemblies were prepared by MicrobesNG against reference sequence *Staphylococcus aureus* RN4220 (taxid: 561307) and pGO1 (NC_012547).

**Southern blotting analysis of evolved pGO1 plasmids and pGO1evol-SaPI hybrids.** All pGO1 evolved plasmids were subjected to diagnostic restriction digestion and Southern blotting analysis to corroborate the sequences obtained via WGS. Briefly, total DNA was extracted from each host strain using the GenElute™ Bacterial Genomic DNA kit (Merck, Germany). 10 μg of total DNA per host strain was digested for 18 h at 37 °C with 30 U of *Bam*HI-HF, *Spe*I-HF and *Pst*I-HF for circular plasmids, or *Bam*HI-HF and *Nru*I-HF for plasmid-SaPI hybrids, in a total volume of 400 μl. All enzymes were obtained from NEB. Subsequently, digestion products were purified via ethanol precipitation, and resuspended in 18 μl ddH$_2$O + 2 μl DNA-loading buffer (10X).

Southern blot analysis was performed essentially as previously described[14]. Briefly, 20 μl of each digested sample was run on 0.7% (w/v) TAE agarose gels at 25 V overnight. The DNA was transferred to a nylon membrane and exposed using a DIG-labelled probe and anti-DIG antibody (1:10,000 (v/v), Roche, product 11093274910), before washing and visualisation. Following visualisation, the blot membrane was washed briefly in ddH$_2$O, stripped (0.2 M NaOH + 0.1% (v/v) SDS) with 2 × 15 min incubations at 37 °C, washed with 2xSSC for 5 mins and re-incubated at 42 °C with prehybridization buffer ready for the next probe. Detection probes for pGO1 or SaPIbov2 DNA were manufactured by PCR using primers listed in Supplementary Table 5. Probe labelling and DNA hybridisation were performed using the protocol for PCR-DIG DNA-labelling and chemiluminescent detection kit (Roche). For analysis of circular plasmids, parallel membranes were probed with *aadA-aph*D (gentamicin-resistance) and *nes* sequences from pGO1; for pGO1evol-SaPIbov2 hybrid elements, parallel membranes were probed with *aadA-aph*D (gentamicin resistance) from pGO1 and *tet*M from SaPIbov2 *bap::tet*M. Uncropped and unprocessed scans of the blots presented in Fig. 4 are available in the accompanying Source Data file.

**Generation of spontaneous rifampicin-resistant mutants.** A spontaneous rifampicin-resistant mutant of strain JP14151 was generated as a recipient strain for mating experiments. Briefly, strain JP14151 was grown overnight in 4 ml TSB at 37 °C, 120 r.p.m. 500 μl of cells were plated onto TSA + 1.7 mM sodium citrate + 1 μg/ml rifampicin and incubated overnight at 37 °C. Colonies representing spontaneous rifampicin-resistant mutants were picked, sub-cultured in TSB + 10 μg/ml rifampicin overnight at 37 °C and plated to give single colonies. 2-3 colonies were tested for stability of the resistance phenotype by culture in TSB without antibiotics for at least 30 generations, followed by subsequent plating on TSA + 1.7 mM sodium citrate + 10 μg/ml rifampicin, yielding strain JP19998. This process was repeated for RN4220, generating the rifampicin-resistant recipient strain JP21894 for examination of plasmid transfer in milk.

**Competitive mating.** Conjugation assays were performed using mixed donor cultures of *S. aureus* SH1000 pGO1 (WT) and *S. aureus* SH1000 with either pGO1evol$_A$, pGO1evol$_F$, pGO1evol$_E$, pGO1evol$_I$, or pGO1evol, and a cadmium- and rifampicin-resistant RN4220 recipient. Overnight cultures of donor and recipient cells were washed with PBS to remove antibiotics and resuspended in TSB without antibiotics. Equal numbers of donor cells (SH1000 containing either pGO1 or pGO1-derivative; 5 × 10$^7$ CFU) were combined, and then the donor mixture was mixed 1:1 with 1 × 10$^8$ CFU RN4220 cadmium- and rifampicin-resistant recipient cells (JP19998). The entire mixture was applied to a sterile 0.45 μm pore nitro-cellulose filter (MF-Millipore™ Membrane Filter, product ID HAWG01300, Merck, UK) placed onto TSA. The mixture was allowed to dry onto the filter before incubating at 37 °C for 24 h. Donor- and recipient-only controls were included in parallel. Following incubation, filters were vortexed vigorously in 1 ml phosphate-buffered saline in order to detach the bacteria from the disks for quantification. To determine plasmid transfer frequencies, transconjugants/transductants, recipients and donors were individually enumerated by plating onto TSA containing appropriate selective antibiotics: total cells, TSA + sodium citrate; total pGO1/pGO1evol donors, 10 μg/ml gentamicin; pGO1-only donors, 10 μg/ml neomycin; transconjugants/transductants, TSA citrate + 10 μg/ml gentamicin, 100 μM CdCl$_2$, and 10 μg/ml rifampicin. Plasmid transfer frequency was defined as the number of pGO1/pGO1evol transconjugants/transductants obtained per donor after 24 h. Colonies carrying the evolved plasmids are sensitive to neomycin, whereas pGO1 encodes the neomycin resistance gene *aadD* within the integrated pUB110 plasmid. Differentiation of pGO1 and pGO1evol was conducted by replica plating randomly picked colonies onto TSA citrate + 10 μg/ml gentamicin and TSA citrate + 10 μg/ml neomycin to determine the antibiotic susceptibility profile of the colony.

Neomycin-sensitive colonies obtained from matings between donor strains carrying pGO1 and pGO1evol$_A$, were subsequently confirmed to be pGO1evol$_A$ by PCR using primer pairs: pGO1_01-F + pGO1_01-R, producing an 800 bp product from shared region of pGO1/pGO1evol$_A$; and pGO1-Frag21F + pGO1-Frag21-R, which produce a 2.5 kb product from pGO1 but no product from pGO1evol$_A$ due to the loss of the pSK639-integrated region.

**Competitive transductions.** From separate overnight cultures, equal numbers of *S. aureus* donor cells carrying pGO1 or pGO1evol were sub-cultured into TSB supplemented with 10 μg/ml gentamicin and grown at 37 °C, 120 r.p.m. to OD$_{540}$ 0.24 ± 0.02. Prophages were induced with 2 μg/ml mitomycin C and incubated at 30 °C, 80 rpm until completely lysed, before filtering with a 0.2 μm syringe filter (Sartorius). 100 μl of the resulting lysate was used to transduce 1 ml RN4220 recipient cells as described above, with selection on TSA plates supplemented with 1.7 mM sodium citrate + 10 μg/ml gentamicin. Plates were incubated for 18-20 h at 37 °C and the colonies were enumerated. Differentiation of pGO1 and pGO1evol was conducted by replica plating colonies onto TSA citrate + 10 μg/ml gentamicin and TSA citrate + 5 μg/ml neomycin to determine the antibiotic susceptibility profile of the colony, with subsequent confirmation by PCR using primer pairs: pGO1_01-F + pGO1_01-R (shared region) and pGO1-1F + pGO1_42317-R, which binds across the deleted region of pGO1evol to give a 1.7 kb product from pGO1evol and a ~27 kb product from pGO1.

**Intra- and intergeneric plasmid transfer.** From overnight cultures, *S. aureus* donor strains (JP18293, JP18406, JP18408 and JP18867 for pC221; JP18292, JP18410, JP18413 and JP18694 for pI258) were sub-cultured into TSB, with appropriate selective antibiotics, and grown at 37 °C, 120 r.p.m. to OD$_{540}$ 0.20 ± 0.02. Prophages were induced by addition of ciprofloxacin (0.6 μg/ml final concentration) and incubated at 30 °C, 80 r.p.m. until completely lysed, before filtering with a 0.2 μm syringe filter (Sartorius). For transductions, 100 μl of phage lysate was used to infect 1 ml of recipient strains (RN4220, JP1220, JP831, or JP7422) at OD$_{540}$ 1.4, supplemented with 4.4 mM CaCl$_2$. After static incubation for 20 min at 37 °C, 3 ml TTA (TSA top agar; 3 g/100 ml TSB, Oxoid, plus 0.75 g/100 ml agar, Formedium) was added to each transduction reaction and the entire contents were poured onto a TSA plate supplemented with 1.7 mM sodium citrate + 10 μg/ml chloramphenicol (pC221) or 100 μM CdCl$_2$ (pI258). Plates were incubated at 37 °C for 24 h for RN4220 and JP1220 recipient strains, 48 h for JP7422, or 72 h for JP831, and the colonies were enumerated.

To confirm the presence of each plasmid in the recipient strains, plasmids were extracted from transductant colonies using a QIAprep Spin Miniprep kit (Qiagen) and subjected to diagnostic restriction digestion analysis. For pC221 candidates, 500 ng of plasmid DNA was digested for 18 h at 37 °C with 10 U of *Xba*I-HF, yielding an expected banding pattern of two fragments at 2.4 kb and 2.1 kb. For pI258 candidates, 300 ng of plasmid DNA was digested for 18 h at 37 °C with 10 U of *Nhe*I-HF and *Eco*RI-HF, yielding an expected banding pattern of seven fragments, at 7.6 kb, 7.2 kb, 5.6 kb, 5.4 kb, 1.4 kb, 0.9 kb and 0.8 kb. All enzymes were obtained from NEB.

**Plasmid transfer in milk.** 2 × 10$^7$ CFUs of donor (JP21924 or JP21925) and recipient (JP21894) strains were grown independently in 10 ml of UHT milk containing 0.6 μg/ml of ciprofloxacin. This sub-inhibitory concentration is represented within the reported concentration range of ciprofloxacin found in dairy cow milk following treatment with the related fluoroquinolone antibiotic, enrofloxacin[57]. Milk cultures were incubated for 6 h at 32 °C with shaking (80 r.p.m.). Subsequently, the same volume (0.5 ml) of the recipient and donor strain were incubated together for 20 min at 37 °C. Afterwards, 3 ml of TSA top agar was added to the mixture, and after gentle mixing, the mixture was poured on TSA agar media containing 1.7 mM sodium citrate, 10 μg/ml rifampicin and 10 μg/ml chloramphenicol. Plates were incubated at 37 °C for 48 h and colonies were counted. Subsequently, colonies were plated on TSA agar containing 3 μg/ml tetracycline to confirm that the transductants had the background of the recipient strain (recipient strain is sensitive to tetracycline, as opposed to the donor strains).

**Bioinformatics analysis.** We downloaded 13,169 complete bacterial genomes from NCBI (Assembly level: complete, downloaded on 15/02/2019). We used the algorithm phiSpy to identify prophages in the bacterial chromosomes[58]. We scanned a total of 8903 genomes, after filtering out genomes which phiSpy could not analyse.

To detect SaPIs in *S. aureus* genomes, we performed BlastP searches using the type I–V SaPIs integrase sequences[55] (sequence identity > 90% and an e-value > 1E-100). The distribution of plasmids, phages and SaPIs in *S. aureus* genomes is presented in Supplementary Data 2. We used CONJScan[20], a MacSyFinder module, to identify conjugative systems in the 243 plasmids found in *S. aureus* genomes.

**Statistical analyses.** Data were tested for normality using the Shapiro–Wilk normality test, then analysed, as indicated in the figure legends, using *t*-tests (two-tailed) or ANOVA (one-way) with Tukey *post hoc* tests for normally-distributed data, or Mann-Whitney (two-tailed) or Kruskal–Wallis with Dunn's *post hoc* tests

for non-parametric data, and Chi-squared tests. All analyses were performed using Graphpad Prism 9 software and R (v. 3.4.2).

**Reporting summary**. Further information on research design is available in the Nature Research Reporting Summary linked to this article.

## Data availability

The source data underlying Figs. 1, 2, 3, 5, 6, and Supplementary Tables 1, 2 and 3 are provided as a Source Data file. Source data are provided with this paper.

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

## Acknowledgements

This work was supported by grants MR/M003876/1, MR/V000772/1 and MR/S00940X/1 from the Medical Research Council (UK), BB/N002873/1, BB/V002376/1 and BB/S003835/1 from the Biotechnology and Biological Sciences Research Council (BBSRC, UK), ERC-ADG-2014 Proposal no. 670932 Dut-signal (from EU), and Wellcome Trust 201531/Z/16/Z to J.R.P.

## Author contributions

J.R.P. conceived the study; S.H., A.S.M., M.T.R., J.C. and A.F.-D. conducted the experiments; S.H., A.S.M., M.T.R., J.C. and A.F.-D., J.C., C.U., R.C.M. and J.R.P. analysed the data. S.H. and J.R.P. wrote the manuscript.

## Competing interests

The authors declare no competing interests.
