## [Peer Review File · Nature Communications]

Staphylococcal phages and pathogenicity islands drive plasmid evolutionREVIEWER COMMENTS

Reviewer #1 (Remarks to the Author):

In this manuscript Hymphrey et al explore the relationship between plasmid size and their horizontal transfer through generalised transduction by temperate phage and SaPIs in *Staphylococcus aureus*.

The authors show that plasmids can be moved between bacterial cells of the same species and different species under laboratory conditions. The transduction efficiencies vary for 3 plasmids that have different sizes, and transduction of large plasmids can lead to selection for recombinants that are smaller and transduced more efficiently. This suggests that plasmid size is an important determinant of transduction efficacy. Finally, the authors carry out bioinformatics analyses to understand whether phage and SaPI-mediated transduction is likely to be a frequent event in nature.

The idea that non-mobile plasmids can spread horizontally by phage-mediated transduction is well established (for example: Orbach, M. J. & Jackson, E. N. Transfer of chimeric plasmids among *Salmonella typhimurium* strains by P22 transduction. *J. Bacteriol.* 149, 985–994 (1982); Deichelbohrer, I., Alonso, J. C., Luder, G. & Trautner, T. A. Plasmid transduction by *Bacillus subtilis* bacteriophage SPP1: effects of DNA homology between plasmid and bacteriophage. *J. Bacteriol.* 162, 1238–1243 (1985); Novick, R. P., Edelman, I. & Lofdahl, S. Small *Staphylococcus aureus* plasmids are transduced as linear multimers that are formed and resolved by replicative processes. *J. Mol. Biol.* 192, 209–220 (1986)). The novelty of this paper is that it may offer an explanation for the observed size distribution of plasmids, since SAPIs and phage can carry distinct plasmid sizes in their capsid. I think some more work needs to be done to further strengthen this point. This study has the potential to make an important advance to our knowledge and understanding of the selective forces that drive plasmid size evolution and their horizontal spread, which will appeal to the broad readership of *Nature Communications*.

Major points:

1- The authors state that phage- and PICI-mediated generalised transduction are the driving forces for intra- and inter-species transfer of non-transmissible plasmids. I agree that their data demonstrate that phage and SAPIs can transduce plasmids, but how important a contribution this process makes to the overall frequencies of intra- and inter-species plasmid transfer in natural environments has not been quantified (and it seems extremely hard to quantify this). I suggest the authors tone down this statement.

2- The authors also state that phage- and PICI-mediated generalised transduction are the driving forces responsible for the size distribution observed in non-conjugative plasmids. This is supported by variation in the observed transduction efficacies of 3 plasmids with different sizes. In addition, under lab conditions, and when selecting for transduction events, the authors observe recombination events that lead to a size reduction of a conjugative plasmid that is critical for its transduction. But how important this is in nature will depend amongst others on how much selection there is for horizontal transmission of a plasmid, whether transduction is the only mechanism that allows for this (also see comment 1), and other selection forces acting on the evolution of plasmid size. The relative strength of these different selection pressures cannot be inferred from this study. I suggest the authors tone down this statement.

3- The variation in transduction efficacies of the 3 plasmids correlates with their sizes. However, plasmid sequence can also play a role (see e.g. Varble et al *Nature Microbiology* 2019, PMID: 30886355). To strengthen the argument that the observed variation in transduction efficiency is driven by variation in size, the authors should determine transduction efficiencies for a larger number of small, intermediate and large plasmids.

4- Related to point 3: The reason why plasmid size matters is that the size of the SAPI and phage capsids varies. This is presented as a given and I suppose this has been characterised in detail in previous work from the same group for the variants studied here? Can the authors add some

more details on this in the text, with references ? Alternatively, add data showing the different capsid sizes for the different elements used in this study, and the amount of DNA they can contain.

5- The authors hypothesise that pC221 packaging by the SaPI-encoded TerSS is more efficient than that promoted by the phage-encoded TerSP. The authors explore this with TerS KO mutants. However, the appropriate way to experimentally test this idea is to swap phage and SAPI TerS sequences, and measure the effect this has on transduction efficiencies.

6- L99: "Since non-transmissible and mobilisable plasmids usually encode antibiotic resistance genes, our results will involve PICIs and phages in the emergence of multi-resistant clones." This is an important point, that can be addressed more carefully in the bioinformatics analysis. Are multi-resistant clones enriched for prophages and/or PICIs ? What is the relationship with their plasmid content ? Are the plasmids that confer antibiotics resistance in *S. aureus* usually non-mobile, or is resistance usually due to conjugative plasmids? Can the authors run a model to explore the interactions between SAPI, phage, plasmids and resistance in greater depth?

7- The data in Figure 8 are fascinating, but deeper analysis of the data is needed to link the plasmid/prophage co-occurrence data to transduction events. As it stands, the correlations may be due to variation in host defence (some hosts being very promiscuous, thus leading to accumulation of prophages and plasmids, others being resistant to everything) and/or phage-encoded counter-defences. One analysis that could be done to understand if the patterns are likely the result of phage-mediated transduction of plasmids is to examine if there is variation in the strength of the correlation in panel D when looking at plasmids of different sizes, and when making a distinction between non-mobile and conjugative plasmids?

Minor points:

1- L332: "Phage and PICI-mediated transfer occurs in nature." Here, the authors present an experiment in a test tube using milk as the growth medium. Please change the title and text to reflect this (e.g. Phage and PICI-mediated transfer occurs in milk). Extrapolating these results to 'nature' is a bit of a stretch.

2- L357: Note that any cell that resists lytic infection cycles will be transduced with higher efficiency. For example, see Watson et al mBio 2018. PMID: 29440578

3- For completeness, I would like the authors to present transduction data for (1) 80a lysogen carrying SaPI-1 *detIa-cpmAB* and plasmid pC221 (the bigger capsids should not affect transduction efficiencies of this small plasmid) and (2) 80a::SaPI-1 *cpmAB* lysogen carrying SaPI1 and plasmid PI258 (a small decrease in transduction efficiency is expected, as for the 80a + SAPI-1).

4- Please provide an overview of the size distributions of plasmids in SAPI+ and SAPI- bacteria in Fig. 7 (which currently only shows the binary small/large classification), as in Fig. S1.

Edze Westra

Reviewer #2 (Remarks to the Author):

In the manuscript the authors investigate the role of phages and PICIs in plasmid transfer. The study starts from the observation of two distinct peaks in the size distribution of non-conjugative plasmids that coincide with the average sizes of phages and PICIs. From there the authors establish a model organism, *S. aureus*, where the rest of experiments will be performed and three natural plasmids of various sizes. They tested the effect that SaPIs, with the help of phage 80 α , has on plasmid transfer. They found that SaPIs affect plasmid transfer and promote it depending on the

plasmid and SaPIs sizes, due to the relation between capsid and plasmid size. Furthermore, in evolution experiments, they saw that the limitation in capsid packaging capacity drives the plasmid remodeling and size reduction to make them compatible with packaging by employing different strategies. These structural modifications result sometimes in the loss of their conjugation capacity. Next, they determined the phage mediated plasmid transfer between species and in milk (as a proxy for natural environment). Finally, surveying the databases they found high small plasmid presence in SaPIs genomes.

General comments

The study overall is well designed and the results quite clear, however, not all findings are entirely new and several of the conclusions are stated as general findings also when these are not.

For example, the whole manuscript (like in line 101) and title, can it be said that phages are the drivers of antibiotic resistance spread? Because as it is seen in the data, many of the transferred plasmid are small hence they are typically depleted from antibiotic genes. Also, the experiments show that many times the reduction in plasmid size occurs via loss of the antibiotic resistance genes. So, I feel that more than antibiotic resistance, they drive spread of just plasmids... A more warrant title for this study would be along the lines of: Phages-mediated transfer constrains plasmid size in *S. aureus*.

Is there an upper limit in the size of the phage/SaPI capsids? Are capsid sizes bigger than 15-40kb found in nature? Maybe in some mutant or something? Just being curious...

A general aesthetics comment, I would homogenize the font type used for the Figures and maybe use a thicker width for the letter so they are more readable, like the one for Fig 7.

Introduction and abstract

I find the abstract too descriptive with the initial finding and background information and it doesn't contain much information about the actual study. Again, it is super general in comparison to the results, which are specific for *S. aureus*.

In the introduction, line 89, I would briefly describe why generalized transduction, in contrast to other modes of transductions, is important for plasmid spread.

Methods

Why is the generation of spontaneous rifampicin mutants the second thing explained in methods? When as far as I can see if not mentioned in the main text until one of the last results sections. It disrupts the reading of the methods in my opinion

Why was the antibiotic ciprofloxacin specifically chosen for the prophage induction? Is there any reason behind it? Is this antibiotic more found in natural settings, like in milk? Or was it just convenience.

Prophage inference – I'm not sure that PhySpy is the state of the art these days and the absence of any threshold makes it look like a "quick and dirty analysis". It is usually more prudent to use the intersection of two different tools (or golden rule of three tools). This is specifically relevant for small (ie short) prophages (that may be often non-functional, see 10.1073/pnas.1405336111).

Results

In general results are clearly indicated and showed in correspondent figures.

In the abstract, introduction and in the study of Smillie et al., the peaks found in plasmid sizes correspond to that of phages and PICIs. In the results from *S. aureus* of Fig S1, the peaks are somehow always smaller, although in the reported range. Is there an explanation for this? Something related to the species? Just wondering

Figures 1,2,3,5 – if I get it right from the legend, the sample size here is 3 (n=3?), with some sets having a sample size of 1 (due to sinking below the detection limit). The current presentation glosses that issue quite well. I recommend switching to boxplots and showing the raw data points on top. Along the same lines, all those figures include statistical tests and an odd number of significance values. I find it difficult to believe that the limited sample size enables a robust statistical analysis (not to mention the use of a parametric test, i.e., ANOVA). Admittedly, its not clear to me what is actually tested here? Considering the limited statistical power I would recommend to just present the results as are (the conclusions from these experiment are pretty clear).

For Figure 1 results of only RN4220 strain are shown, was the behavior in SH1000 similar? If so, it could be indicated in the legend or results section

Just as a suggestion, maybe in the Results section "SaPIs severely impact plasmid structure and transfer" when presenting results from Fig1, I would explain at the beginning in a sentence the results from the 3 plasmids just stating that for the smaller the size more transfer or something. Also, nothing is really said about SaPIbov2 for plasmid pC221. In general, I feel like this section, although clear in each paragraph and straightforward, has some issues with the order the experiments are presented. It makes the reader go jumping back and forth looking at the figures. Maybe Might be better to explain Fig1 entirely and then go testing the rest of hypothesis with the mutants..

In line 178, after the behavior with SaPIbov2 I would add a Ref to the figure where this is shown (Fig 1B) again because you are jumping back and forth.

Then in Fig3, the usual order of the plasmids is changed, first pI258 and then pC221.

In line 196, the multiple separate experiments mentioned from where the evolved plasmids came from, is there any relationship between the experiments and rearrangements found, in the sense of whether the ones presenting similar mutations came from the same genetic background. Just wondering...

I'm curious about the evolved pGO1evolF and pGO1evolA, that show differences in transfer efficiencies when in competition. Do you have any hypothesis for the reason behind it? Because in the case of pGO1evolA is practically the same as pGO1evolB as far as I saw.

Line 250/262 Plasmid genome size evolution /recombination due to the presence of IS elements – (described as fascinating) – the contribution of IS elements to plasmid genome rearrangements has been studied (and published) quite extensively in the 1970s (e.g., see this review: <https://www.nature.com/articles/263731a0>).

Line 332 – the title of this section should be toned down. ... transfer occurs in milk (not nature). While the experiment in milk is a nice addition to test the boundaries of plasmid-mediated transfer, I think that also here, the conclusions go far beyond the clear observations.

Line 366 – The context of the statistical test is not clear. Looking at Fig. 7, about half of the isolates with SaPIs harbor also small plasmids (50/95), which would be then close to random (i.e., flip of a coin).

Line 371 – the analysis comparing plasmid and phage sizes is extremely superficial and the conclusions far-fetched. Seeing two distributions with a similar shape cannot be used to conclude about causality (in other words, beware of the storks! A new parameter for sex education | Nature <https://www.nature.com/articles/332495a0>)

Discussion

In the databases surveyed, the plasmids you found, where there traces in their genomes of phage

and PICIs sequences? Because that could serve as support for these plasmids found were maybe reduced in size due to transduction capsid size limitations, like in the case of pGO1 hybrids and possibly pACK2.

Lines 427-429 – the authors demonstrate this statement (on mobilizable plasmids) only for *S. aureus* and their conclusions should be state accordingly. They can “suggest” that their finding is more general.

Line 430 – previous studies demonstrated plasmid transfer by phages, also specifically for *S. aureus* – Im quite surprised that studies from R. Pantůček on the topic are not cited in this manuscript.

Line 438 – about the consequences of phage host range to horizontal transfer by transduction, see this publication, based on phylogenetic reconstruction: doi:10.1038/ismej.2016.116

Line 468 – on phages shaping plasmid evolution – this is an exaggerated statement. The findings are specific for *S. aureus* and the effect suggested by the results is likely on plasmid genome size (not, e.g., gene content).

In line 509-512– again, an exaggerated conclusion – considering that phages are not universal for all taxa. The authors do not have results to support that conclusion. Also, I do not fully agree that less genes decreases the vertical transmission. If a plasmid has addiction system it will be transmitted regardless of the number of genes it carries. It will only be less fit according to the host point of view in presence of selection. In other words – I think that the last discussion paragraph is very speculative (of course, it’s up to the authors to write their opinion, yes, a more suggestive language would be prudent).

Reviewer #3 (Remarks to the Author):

The manuscript “Phages and pathogenicity islands drive plasmid evolution an the spread of antibiotic resistance” by Humphrey et al. describes a novel hypothesis to explain the transfer mechanisms as well as evolutionary constraints of a class of plasmids found within the bacterial domain of life. The investigation is timely as it touches on emerging questions, revealed by large genomics analyses that have shown the abundant existence of these so-called non-transmissible plasmids in many bacterial species.

The authors of the study propose that i) transduction from (pro-)phages and phage-inducible chromosomal islands (PICIs) are the main driver supporting the existence and spread of these otherwise non-transmissible plasmids, and ii) that there is an evolutionary tradeoff involved in this mode of transmission, which is imposed by the limitation of capsid sizes that can accomodate size-limited genomic elements. To experimentally support their hypothesis, the authors chose to work with *Staphylococcus aureus*, well-known to harbour plasmids, inducible prophages, and SaPI (PICI) that rely on helper prophages. Indeed, the authors make a compelling demonstration of the interplay between three plasmids, chosen based on their size, and transduction mediated by two different SaPIs and the 80alpha prophage. Furthermore, the authors show that these transduction events extend to a few other species, transferring antibiotic resistance genes.

This is a well written manuscript, with a comprehensive investigation that includes both genetic validation and bioinformatics analysis. The experiments conducted and the results do support the hypothesis presented. Additionally, this has important implications for the study of mechanisms of horizontal gene transfer in bacteria, and also in our understanding of the spread of antibiotic resistance genes.

Major comments:

None.

Minor comments:

- Line 110: reference here your supplementary table with the accession numbers (ST7). You do not mention how you selected the 295 genomes. A quick search on NCBI reveals that there are many more (including complete ones).
- Are multiple copies of pC221 present in the SAPI1 capsids? Or are those small SaPI-sized capsids stable and can accommodate smaller genomes (4.6 kbp vs. 15 kbp)?
- Although I do think the conclusions are otherwise well supported, it appears that you did not treat your lysates with DNase (or at least that is not mentioned in your M&M, only 0.2 μ m size filtration is mentioned). That must mean that abundant amounts of DNA is available outside of the capsids when conducting your main transduction assays to *S. aureus*, *S. epidermidis*, *S. xylosum*, and *L. monocytogenes*. *S. aureus* does have competence mechanism via SigH, and it could be argued that limited amounts of transformation could occur and confound your results – could you clarify whether you think that could be the case or if you ruled that out completely?
- Furthermore, Mitomycin C is also still present in the lysate that you use to transduce your cells. In some species, it has been shown to induce competence.
- Could you explain further how the five evolved plasmids (pGO1evol_a to f) were created? You mention multiple separate experiments but it isn't clear why you would need multiple experiments... or do you mean the replicates of the experiments that led to Figure 1C?
- In line 372, do you mean any plasmids? Or the non-transmissible plasmids?
- In your discussion (431-435) you warn about the impacts of the mechanism you have described in the field of phage therapy. This is very interesting, but as far as I know, we do not use "poorly transducing" phages for therapy – typically, if the genome analysis shows a sign of potential lysogenic lifestyle, the phage is no more a candidate. Do you contend that strictly lytic phages could also be hijacked for the transduction of plasmids as you have described here?

Cédric Lood

REVIEWER COMMENTS

Reviewer #1 (Remarks to the Author):

The idea that non-mobile plasmids can spread horizontally by phage-mediated transduction is well established (for example: Orbach, M. J. & Jackson, E. N. Transfer of chimeric plasmids among *Salmonella typhimurium* strains by P22 transduction. *J. Bacteriol.* 149, 985–994 (1982); Deichelbohrer, I., Alonso, J. C., Luder, G. & Trautner, T. A. Plasmid transduction by *Bacillus subtilis* bacteriophage SPP1: effects of DNA homology between plasmid and bacteriophage. *J. Bacteriol.* 162, 1238–1243 (1985); Novick, R. P., Edelman, I. & Lofdahl, S. Small *Staphylococcus aureus* plasmids are transduced as linear multimers that are formed and resolved by replicative processes. *J. Mol. Biol.* 192, 209–220 (1986)). The novelty of this paper is that it may offer an explanation for the observed size distribution of plasmids, since SAPIs and phage can carry distinct plasmid sizes in their capsid. I think some more work needs to be done to further strengthen this point.

We completely agree with this reviewer in the fact that several papers had demonstrated that phages can mobilise plasmids in the lab. In spite of this, the reality is that the impact that phages have on plasmid transfer and plasmid biology is currently unknown. Thus, many of these plasmids are called non-mobilisable, clearly reinforcing the idea that phages were never considered seriously as elements that could impact on the mobility and biology of these elements.

This study has the potential to make an important advance to our knowledge and understanding of the selective forces that drive plasmid size evolution and their horizontal spread, which will appeal to the broad readership of *Nature Communications*.

Thanks for such nice comments.

Major points:

1.1- The authors state that phage- and PICI-mediated generalised transduction are the driving forces for intra- and inter-species transfer of non-transmissible plasmids. I agree that their data demonstrate that phage and SAPIs can transduce plasmids, but how important a contribution this process makes to the overall frequencies of intra- and inter-species plasmid transfer in natural environments has not been quantified (and it seems extremely hard to quantify this). I suggest the authors tone down this statement.

We have amended this statement in lines 37-39 to read:

*“Here, using *Staphylococcus aureus*, we demonstrate that phages and PICIs can mediate intra- and inter-species plasmid transfer via generalised transduction, potentially contributing to non-transmissible plasmid spread in nature.”*

And have added the word ‘potentially’ to line 433, which now reads:

“...demonstrate that phages and PICIs are potentially key players in promoting both intra- and inter-species plasmid transfers in nature.”

1.2- The authors also state that phage- and PICI-mediated generalised transduction are the driving forces responsible for the size distribution observed in non-conjugative plasmids. This is supported by variation in the observed transduction efficacies of 3 plasmids with different sizes. In addition, under lab conditions, and when selecting for transduction events, the authors observe recombination events that lead to a size reduction of a conjugative plasmid that is critical for its transduction. But how important this is in nature will depend amongst others on how much selection there is for horizontal transmission of a plasmid, whether transduction is the only mechanism that allows for this (also see comment 1), and other selection forces acting on the evolution of plasmid size. The relative strength of these different

selection pressures cannot be inferred from this study. I suggest the authors tone down this statement.

We have attempted to tone down this statement by using more suggestive language.

We have amended the sentence in lines 444-446 to read: *“Our results demonstrate that in S. aureus, mobilisable plasmids can significantly increase their transferability by exploiting phages, PICIs and conjugative plasmid machineries for transfer, highlighting the fascinating strategies open to these plasmids to permit spread in nature.”*

We have amended the sentence in line 484-486 to read: *“More generically, since most of the PICIs described to-date have the ability to promote the formation of PICI-sized capsids^{53,11,12}, it is interesting to speculate that PICIs may also drive the evolution and transfer of small-sized plasmids in species other than S. aureus.”*

We have amended the sentence in line 537 to read: *“This complex interplay between elements ultimately culminates in a potentially important role for phages and SaPIs in dictating the distribution of plasmids in S. aureus on account of mediating their transfer by generalised transduction.”*

1.3- The variation in transduction efficacies of the 3 plasmids correlates with their sizes. However, plasmid sequence can also play a role (see e.g. Varble et al Nature Microbiology 2019, PMID: 30886355). To strengthen the argument that the observed variation in transduction efficiency is driven by variation in size, the authors should determine transduction efficiencies for a larger number of small, intermediate and large plasmids.

We think there is a bit of confusion here. We completely agree that plasmid sequences are important in plasmid transfer, but that was out of the scope of this manuscript. What we propose is that the plasmid size matters in the context of the size of the transducing particles that will mobilise them. The results obtained with the use of the different SaPI mutants, incapable of producing SaPI-sized capsids, or the reduction in size observed in the pGO1 evolved plasmids support this.

1.4- Related to point 3: The reason why plasmid size matters is that the size of the SaPI and phage capsids varies. This is presented as a given and I suppose this has been characterised in detail in previous work from the same group for the variants studied here? Can the authors add some more details on this in the text, with references? Alternatively, add data showing the different capsid sizes for the different elements used in this study, and the amount of DNA they can contain.

We apologise that we did not make this information clear enough in our initial draft of the manuscript. We have added further details of the rationale for the selection of each SaPI element, along with the relevant studies involved in characterising the SaPIs, in lines 146-155. This new section reads:

“SaPI1 was chosen because the process of remodelling helper phage capsids to produce small SaPI capsids (with DNA packaging capacities of ~15 kb) has been particularly well characterised for this phage-SaPI pairing^{16,24}. Following its induction by phage 80α, SaPI1 scavenges phage capsid proteins and utilises two capsid morphogenesis genes (cpmAB) to reconstruct capsids with reduced DNA packaging capacity, at around 33% that of their 80α counterpart²⁴. Conversely, we selected SaPIbov2 as it is atypically large (27 kb) owing to a transposon insertion in its genome²⁵, making it too big to fit into small SaPI capsids. SaPIbov2 circumvents this issue by being a natural cpmB mutant, abolishing capsid remodelling to small SaPI capsids, and thus exclusively packaging into phage-sized capsids (with DNA packaging capacities of ~45 kb)²⁶.”

1.5- The authors hypothesise that pC221 packaging by the SaPI-encoded TerS_S is more efficient than that promoted by the phage-encoded TerS_P. The authors explore this with TerS KO mutants. However, the appropriate way to experimentally test this idea is to swap phage and SaPI TerS sequences, and measure the effect this has on transduction efficiencies.

We respectfully disagree with the reviewer regarding this approach to comparing the TerS_P and TerS_S packaging efficiencies. In phage 80 α , TerS_P is located at the beginning of a long operon encoding the morphogenetic and lysis modules, which contain essential genes encoding phage virion components and proteins involved in packaging and virion particle assembly. The *terS* (*gp40*) and *terL* (*gp41*) genes share an 80bp sequence overlap, prohibiting complete removal of *terS_P* for exchange with *terS_S* without disruption to downstream genes of critical importance for phage (or transducing particle) packaging and assembly. Accordingly, in order to preserve functionality of the operon, the suggested approach would necessitate the creation of chimeric versions of the *terS_S-terS_P* sequences, partially retaining the 3' sequence of *terS_P*, the presence of which may confound the results as we do not know which part of the TerS protein is involved in *pac* (or *pseudo-pac*, in the case of the plasmids) site recognition or whether the residual TerS_P sequence would disrupt formation of the TerS_S multimer and affect its ability to package the plasmids for transduction. Thus, we do not feel that we would be able to swap the *terS* sequences with sufficient confidence to adequately test their effect on plasmid transduction efficiency.

Though we chose not to include it in this manuscript due to space constraints, we did perform a related experiment to further compare the packaging efficiencies of TerS_P and TerS_S for plasmids pC221 and pI258 using inducible plasmids to over-express either the 80 α TerS (unpublished) or SaPI_{bov1} TerS (pJP368 – see doi:10.1111/j.1365-2958.2007.05758.x.) in either the 80 α Δ *terS* pC221 or 80 α Δ *terS* pI258 background. (Note that SaPI_{bov1} and SaPI1 small terminase sequences are highly similar: 95% identical at the nucleotide level and 98% identical at the protein level.) An empty expression plasmid control was also included. Strains carrying the expression plasmids and either pC221 or pI258 were induced at the same time as MC was added to activate the 80 α Δ *terS* prophage, and the resulting lysates were sterile filtered following lysis. pC221 or pI258 transfer from each background was then tested in an RN4220 recipient.

No transfer was observed from the 80 α Δ *terS* pC221 donor background carrying the empty expression plasmid. When TerS_P was overexpressed, pC221 transfer was 4.69 (\pm 0.04) log₁₀ TrU/ml, but when TerS_S was overexpressed, transfer was increased to 5.84 (\pm 0.14) log₁₀ TrU/ml, indicating that SaPI_{bov1} TerS is more efficient than 80 α TerS at packaging plasmid pC221. We used variants of the same expression plasmids with pI258, however we swapped the erythromycin resistance cassette on the inducible plasmids with chloramphenicol resistance in order to enable compatibility with pI258. The results were similar for pI258 to that seen for pC221: no transfer was observed when the empty plasmid was induced, 5.78 (\pm 0.08) log₁₀ TrU/ml were obtained with TerS_P was overexpressed, while 6.29 (\pm 0.09) log₁₀ TrU/ml were obtained when SaPI_{bov1} TerS was overexpressed, indicating that TerS_S was more efficient than TerS_P at packaging the pI258 plasmid for transduction. These results served to substantiate our observations using the *terS* mutants (presented in Fig. 2) that TerS_S is more efficient than TerS_P at plasmid packaging.

1.6- L99: "Since non-transmissible and mobilisable plasmids usually encode antibiotic resistance genes, our results will involve PICs and phages in the emergence of multi-resistant clones." This is an important point, that can be addressed more carefully in the bioinformatics analysis. Are multi-resistant clones enriched for prophages and/or PICs? What is the relationship with their plasmid content? Are the plasmids that confer antibiotics resistance in *S. aureus* usually non-mobile, or is resistance usually due to conjugative plasmids? Can the authors run a model to explore the interactions between SaPI, phage, plasmids and resistance in greater depth?

We thank the reviewer for this insightful comment. One of the main limitations of the large databases containing sequenced genomes is the fact that, in the case of pathogenic species, most of these genomes come from clinically relevant isolates. Due to this bias, most of the *S. aureus* genomes available carry multiple antibiotic resistance genes and it is therefore difficult to try to correlate the multi-resistance genotype with the presence of phages (which are in fact pervasive) or SaPIs.

Antibiotic resistance in *Staphylococcus aureus* can be transmitted both by small and large plasmids. Actually, there are multiple reports of small plasmids carrying AR genes (e.g. doi.org/10.3389/fmicb.2018.02063). Of course, “large” conjugative plasmids carrying AR genes have also been described (although to a much lesser extent than in other bacterial groups such as enterobacteria), and have been studied in detail. In *S. aureus* there seems to be a bias toward small plasmids compared to other species/families (see doi:10.1128/microbiolspec.PLAS-0039-2014). In fact, looking at figure S1 or to the new panel in figure 7 one can see how nearly all plasmids in *S. aureus* are below 45 kb, which is the peak of the unimodal distribution of phage sizes.

Regarding plasmid mobility, dividing *S. aureus* plasmids between mobile/non-mobile bioinformatically is difficult. Recent reports in *S. aureus* indicate that many of the plasmids previously considered as non-mobile (not carrying conjugative type IV secretion systems, and not even relaxases) are in fact able to be transferred through conjugation due to the presence of a simple origin of transfer (see DOI: 10.1093/nar/gkv755 and 10.1080/2159256X.2016.1208317). Therefore, the classic bioinformatic approaches used to categorize plasmids into conjugative / mobilizable / non-transmissible may not be as reliable as previously thought. That’s why, even though we considered performing this type of analysis when we started the project, we subsequently declined taking this approach.

In summary, given that we have not been able to investigate interactions between SAPI, phage, plasmids and resistance in greater depth as suggested by the reviewer, we have toned down the sentence in line 99 (now line 110) to: “*Since non-transmissible and mobilisable plasmids usually encode antibiotic resistance genes, our results suggest that PICIs and phages could play a relevant role in the emergence of multi-resistant clones*”.

1.7- The data in Figure 8 are fascinating, but deeper analysis of the data is needed to link the plasmid/prophage co-occurrence data to transduction events. As it stands, the correlations may be due to variation in host defence (some hosts being very promiscuous, thus leading to accumulation of prophages and plasmids, others being resistant to everything) and/or phage-encoded counter-defences. One analysis that could be done to understand if the patterns are likely the result of phage-mediated transduction of plasmids is to examine if there is variation in the strength of the correlation in panel D when looking at plasmids of different sizes, and when making a distinction between non-mobile and conjugative plasmids?

We are happy to hear that the reviewer liked Figure 8 and we thank him for this great idea. As suggested by the reviewer, we have tried to examine if there is variation in the strength of the correlation in panel D when looking at plasmids of different sizes. However, there is one problem with performing this analysis, which is the fact that in panel D we look at frequency of plasmid-bearing genomes, and most of the genomes carry more than one plasmid. Most of the genomes carry both large and small plasmids at the same time, actually, so we cannot categorize them as “carrying large” or “carrying small” plasmids. To try to circumvent this limitation we thought of an alternative analysis, which is looking at the distribution of plasmid sizes within each group of genomes according to the number of phages (0, 1, 2...>9, the groups in panel D, x-axis). The idea is that those genomes enriched in phages may carry a higher proportion of small plasmids. We ran the analysis and these are the results:

In this plot we represent all the groups together:

We observed how the bimodal distribution and the separation between the peaks seems to increase from the genomes with no prophages to those genomes carrying increasing number of prophages (at least up to 4), but we did not observe an evident increase in the proportion of small plasmids. Therefore, as these results are not conclusive, we decided not to include them in the manuscript. However, we now clarify that the observed results could be the result of differences in host defense systems that may lead to accumulation (or absence) of multiple phages/plasmids in the same genomes and have added the following statement in line 401: *“However, other factors apart from phage-mediated mobilization, such as for example differences in host defense systems, may help to explain the co-occurrence of phages and plasmids in genomes, and further work will be required to confirm our hypothesis.”*

Regarding the distinction between non-mobile and conjugative plasmids, as we explained in the answer to the previous question, making a distinction between non-mobile and conjugative plasmids is not evident in *S. aureus* so we couldn't perform this analysis.

Minor points:

1.8- L332: "Phage and PICI-mediated transfer occurs in nature." Here, the authors present an experiment in a test tube using milk as the growth medium. Please change the title and text to reflect this (e.g. Phage and PICI-mediated transfer occurs in milk). Extrapolating these results to 'nature' is a bit of a stretch.

We have amended the title of this section to read: *“Phage- and PICI-mediated plasmid transfer occurs in milk.”*

We have also amended line 349-350 to read: *“Accordingly, milk represents a more natural model environment for examining phage- or PICI-mediated plasmid transfer.”*

1.9- L357: Note that any cell that resists lytic infection cycles will be transduced with higher efficiency. For example, see Watson et al mBio 2018. PMID: 29440578

We have added the following sentence (including the reference) as a precursor in line 371-374:

“Bacterial cells that can resist lytic phage activity are likely to be transduced more efficiently than those susceptible to phage-mediated killing³⁵. Since prophages protect their host cell from infecting phages, so-called lysogenic immunity, transduction primarily occurs in cells that have been previously lysogenised³⁶.”

1.10- For completeness, I would like the authors to present transduction data for (1) 80a lysogen carrying SaPI-1 Δ cpmAB and plasmid pC221 (the bigger capsids should not affect transduction efficiencies of this small plasmid) and (2) 80a::SaPI-1 cpmAB lysogen carrying SaPI1 and plasmid pI258 (a small decrease in transduction efficiency is expected, as for the 80a + SAPI-1).

As requested, we have added this data in Figure 3. (1) Curiously, our data indicated that plasmid pC221 transfer was further enhanced in the 80 α SaPI1 Δ cpmAB donor compared to the 80 α SaPI1 donor, however further scrutiny revealed that in the cpmAB mutant background, the total number of transducing particles (phage + SaPI particles) was ~1.4 logs higher than in the 80 α SaPI1 WT background (because the SaPI can no longer severely restrict phage capsid size). This difference correlated with the 1.3 log difference observed for pC221 transfer between these two strains, suggesting that the increase in plasmid transfer is due to the greater availability of transducing particles in the 80 α SaPI1 Δ cpmAB donor. Unfortunately, we are not able to universally normalise the plasmid transduction rate by the total number of transducing particles for all donor backgrounds because the number of transducing particles cannot be determined for the 80 α ::SaPI1cpmAB mutant, so we have presented the data as the log₁₀-transformed absolute TrU per ml of culture. We will provide all raw data titre values for full transparency along with this manuscript upon submission. (2) As predicted by the reviewer, plasmid pI258 transfer was indeed slightly further decreased in the 80 α ::SaPI1cpmAB SaPI1 background. Importantly, changes in the total number of transducing particles did not correlate with the differences observed in pI258 transduction between the strains, suggesting that the effects observed for this plasmid were entirely due to changes in transducing particle capsid size restricting or permitting pI258 packaging and transfer.

1.11- Please provide an overview of the size distributions of plasmids in SAPI+ and SAPI- bacteria in Fig. 7 (which currently only shows the binary small/large classification), as in Fig. S1.

We have included a new panel in figure 7 showing of the size distributions of plasmids in SAPI+ and SAPI- bacteria, as suggested.

Reviewer #2 (Remarks to the Author):

General comments

2.1 The study overall is well designed and the results quite clear, however, not all findings are entirely new and several of the conclusions are stated as general findings also when these are not.

For example, the whole manuscript (like in line 101) and title, can it be said that phages are the drivers of antibiotic resistance spread? Because as it is seen in the data, many of the transferred plasmid are small hence they are typically depleted from antibiotic genes. Also, the experiments show that many times the reduction in plasmid size occurs via loss of the antibiotic resistance genes. So, I feel that more than antibiotic resistance, they drive spread of just plasmids... A more warrant title for this study would be along the lines of: Phages-mediated transfer constrains plasmid size in *S. aureus*.

We thank the reviewer for the comment, but I am afraid this observation is not correct. First, many small plasmids carry antibiotic resistance genes in *S. aureus* (see doi.org/10.3389/fmicb.2018.02063). Second, nearly all plasmids in *S. aureus* are below 45 kb (Figure 7), which is the peak of the unimodal distribution of phage sizes, and therefore most of them can be, in principle, transduced.

2.2 Is there an upper limit in the size of the phage/SaPI capsids? Are capsid sizes bigger than 15–40kb found in nature? Maybe in some mutant or something? Just being curious...

SaPIs are entirely dependent on their helper phage providing the capsid proteins required for production of the SaPI virion, hence the upper limit for SaPI capsids is entirely dependent on the size of the helper phage capsid. This is demonstrated nicely in the case of SaPI_{bov2}, where this unusually large SaPI does not engage in capsid remodelling (as its genome would be too large to fit in small SaPI capsids) and instead uses the helper phage capsids to package its genome.

The majority of well-characterised *S. aureus* *Siphoviridae* phages have genomes of around 45kb in size, however, there have been some reports in the literature of large and jumbo (>200 kb genome size) *S. aureus* phages, typically belonging to the *Myoviridae* family (see <https://doi.org/10.1007/s00284-021-02395-y>; <https://doi.org/10.1101/2020.12.14.422802>; <https://doi.org/10.1038/ismej.2014.29>), which exhibit broad host ranges and have demonstrated an ability to transduce plasmids and chromosomal DNA between *Staphylococci* via GT. To the best of our knowledge, however, it is unclear how widespread such phages might be among natural *S. aureus* populations, which makes analysing their contribution to plasmid transmission (and potential plasmid evolution driven by physical limitations of capsid packaging) challenging.

2.3 A general aesthetics comment, I would homogenize the font type used for the Figures and maybe use a thicker width for the letter so they are more readable, like the one for Fig 7.

We have attempted to improve the aesthetics by making the font type used in the figures bolder to improve readability.

Introduction and abstract

2.4 I find the abstract too descriptive with the initial finding and background information and it doesn't contain much information about the actual study. Again, it is super general in comparison to the results, which are specific for *S. aureus*.

We have rewritten the abstract to make it more specific to *S. aureus* and to state the main findings of our results more explicitly.

2.5 In the introduction, line 89, I would briefly describe why generalized transduction, in contrast to other modes of transductions, is important for plasmid spread.

We have added the following description of generalised transduction in lines 96-101:

“GT is the process by which phages or PICs that use the headful mechanism of packaging mobilise either chromosomal or plasmid DNA from one bacterium to another. The process is initiated by either the phage- or SaPI-encoded small Terminase subunit (TerS) which occasionally recognises, with low frequency, pac site homologs (also called pseudo-pac sites) in host chromosomal or plasmid DNA, initiating packaging into the phage or SaPI capsid to form transducing particles.”

Methods

2.6 Why is the generation of spontaneous rifampicin mutants the second thing explained in methods? When as far as I can see if not mentioned in the main text until one of the last results sections. It disrupts the reading of the methods in my opinion

We have now moved this section so that it is located between the *Southern blotting analysis of evolved pGO1 plasmids and pGO1evol-SaPI hybrids* and *Competitive Mating* sections of the methods.

2.7 Why was the antibiotic ciprofloxacin specifically chosen for the prophage induction? Is there any reason behind it? Is this antibiotic more found in natural settings, like in milk? Or was it just convenience.

We opted to use ciprofloxacin at sub-inhibitory concentrations in the milk model to attempt to reflect a more natural stimulus for prophage activation than Mitomycin C. Ciprofloxacin is a fluoroquinolone antibiotic that is used clinically in humans, functioning by inactivating bacterial DNA replication via topoisomerase inhibition. At sub-inhibitory concentrations in *S. aureus*, ciprofloxacin activates prophages present within the cell, most likely via the activity of the bacterial SOS response (<https://doi.org/10.1128/AAC.50.1.171-177.2006>). Though ciprofloxacin itself is not commonly used in animal husbandry, it is produced as a primary, active metabolite in animals following the breakdown of another fluoroquinolone antibiotic, enrofloxacin. The sub-inhibitory 0.6 µg/ml concentration of ciprofloxacin used here is representative of concentrations reported to be present in cow's milk up to ~18-20h post-administration of enrofloxacin (<https://doi.org/10.1016/j.rvsc.2009.12.019>).

We have added the following to the results and materials and methods sections relating to this experiment to explain our rationale for the use of ciprofloxacin at this concentration:

Results (line 355): *“Donor and recipient strains were individually incubated in milk containing subinhibitory concentrations of the antibiotic ciprofloxacin to activate the prophages.”*

Materials and Methods (lines 673-675): *“This sub-inhibitory concentration is represented within the reported concentration range of ciprofloxacin found in dairy cow milk following treatment with the related fluoroquinolone antibiotic, enrofloxacin...”*

2.8 Prophage inference – Im not sure that PhySpy is the state of the art these days and the absence of any threshold makes it look like a “quick and dirty analysis”. It is usually more prudent to use the intersection of two different tools (or golden rule of three tools). This is specifically relevant for small (ie short) prophages (that may be often non-functional, see 10.1073/pnas.1405336111).

We do not agree with the analysis being “quick and dirty”. It is true that using two or three different tools may have been better, but PhySpy is an updated and reliable tool (see <https://github.com/linsalrob/PhiSpy>). Regarding the “absence of any threshold”, PhySpy by default uses “strict mode”, where it looks for two or more genes that are likely to belong to a phage in each prophage region. Therefore, we consider that given the goal of our analysis (a wide scan of phages across thousands of genomes) even if a small fraction of the hits are false positives the results are still reliable and robust.

Results

In general results are clearly indicated and showed in correspondent figures.

2.9 In the abstract, introduction and in the study of Smillie et al., the peaks found in plasmid sizes correspond to that of phages and PICs. In the results from *S. aureus* of Fig S1, the peaks are somehow always smaller, although in the reported range. Is there an explanation for this? Something related to the species? Just wondering

Plasmid size correlates with chromosome size in bacterial species (this result is also shown in Smillie et al.). Therefore, although the bimodal distribution of plasmid sizes holds across bacterial phylogeny (see figure 1b in <https://doi.org/10.1038/s41579-020-00497-1>), the locations of the peaks are different for different species. Since *S. aureus* have small genomes, the peaks are also smaller than in the general distribution.

2.10 Figures 1,2,3,5 – if I get it right from the legend, the sample size here is 3 (n=3?), with some sets having a sample size of 1 (due to sinking below the detection limit). The current presentation glosses that issue quite well. I recommend switching to boxplots and showing the raw data points on top. Along the same lines, all those figures include statistical tests and an odd number of significance values. I find it difficult to believe that the limited sample size enables a robust statistical analysis (not to mention the use of a parametric test, i.e., ANOVA). Admittedly, its not clear to me what is actually tested here? Considering the limited statistical power I would recommend to just present the results as are (the conclusions from these experiment are pretty clear).

We have taken the reviewer’s criticisms on board and have remade the indicated figures as plots with the median (bold bar) and range presented for each group, with individual data points overlaid. Regarding the ‘odd number of significance values’, the significance values stated in the figure legends are the exact p values obtained for each comparison. While we do accept the reviewer’s point regarding limited statistical power for the data sets (n=3), we believe that ANOVA with Tukey post-tests is appropriate for analysis of the data presented in Figure 1, Figure 2a, Figure 3, as these data sets pass the Shapiro-Wilk normality test ($p > 0.05$). We have revisited the remaining figures (2b and 5) and have performed non-parametric tests on these data, the results of which are now reported in their respective figure legends. We have updated the methods section of the manuscript to reflect these changes:

Data were tested for normality using the Shapiro-Wilk normality test, then analysed, as indicated in the figure legends, using t-tests (two-tailed) or ANOVA (one-way) with Tukey post hoc tests for normally-distributed data, or Mann-Whitney (two-tailed) or Kruskal Wallis with Dunn’s post hoc tests for non-parametric data, and Chi-squared tests. All analyses were performed using Graphpad Prism 9 software and R (v. 3.4.2).

2.11 For Figure 1 results of only RN4220 strain are shown, was the behavior in SH1000 similar? If so, it could be indicated in the legend or results section

For all experiments, transductions were performed with RN4220 as the recipient strain. This is consistent throughout the paper unless otherwise specified. The parental donor strains differ

between RN4220 and SH1000 in Fig.1: for pC221 and pI258, their original parental strain was in an RN4220 background, hence derivatives from these parental strains, i.e. 80 α lysogens +/- SaPIs were all prepared in the RN4220 background. Conversely, the pGO1 parental strain was the SH1000 background, thus all its derivative strains are in the SH1000 background because of the difficulty involved in moving this plasmid to a clean (i.e. without selectable antibiotic markers) RN4220 background. A direct comparison between the transduction efficiencies of RN4220 and SH1000 donors was not made in this experiment, however separate experiments comparing transduction of the pGO1 variants from each donor background have indicated no discernible differences between the strains.

2.12 Just as a suggestion, maybe in the Results section “SaPIs severely impact plasmid structure and transfer” when presenting results from Fig1, I would explain at the beginning in a sentence the results from the 3 plasmids just stating that for the smaller the size more transfer or something. Also, nothing is really said about SaPI_{bov2} for plasmid pC221. In general, I feel like this section, although clear in each paragraph and straightforward, has some issues with the order the experiments are presented. It makes the reader go jumping back and forth looking at the figures. Maybe Might be better to explain Fig1 entirely and then go testing the rest of hypothesis with the mutants...

We have split up this section into three sub-sections to make it clearer for the reader. The first sub-section deals with the experiments presented in Figure 1, the second refers to the experiments presented in Figure 2, and the third section refers to the experiments in Figure 3.

2.13 In line 178, after the behavior with SaPI_{bov2} I would add a Ref to the figure where this is shown (Fig 1B) again because you are jumping back and forth.

Added.

2.14 Then in Fig3, the usual order of the plasmids is changed, first pI258 and then pC221.

We have swapped the order of the figures to show the results for pC221 as Fig. 3a and pI258 as Fig. 3b so that they are consistent with previous figures.

2.15 In line 196, the multiple separate experiments mentioned from where the evolved plasmids came from, is there any relationship between the experiments and rearrangements found, in the sense of whether the ones presenting similar mutations came from the same genetic background. Just wondering...

There are links between some of the variants obtained and genetic background, however this is not true for all of the pGO1_{evol} variants. For example, variants pGO1_{evolA} and pGO1_{evolB} were obtained from early pilot studies from the same SH1000 pGO1 80 α SaPI_{bov2} background, though from different transducing lysates generated from distinct biological replicates. Following a change in lab personnel, and since the previous result was unexpected, we remade the donor strain following the same procedure as was used to generate the original version (yielding SH1000 pGO1 80 α SaPI_{bov2} = JP19996) to ensure that the pGO1_{evolA} and B variants were not artefacts. The transduction experiment was then repeated (generating the data presented in Fig. 1C) and yielded three different pGO1 variants: pGO1_{evolC} (SaPI_{bov2}::pGO1 hybrid) obtained from experimental repeat 2, pGO1_{evolF} obtained from repeat 2, and pGO1_{evolJ} (SaPI_{bov2}::pGO1 hybrid) obtained from repeat 3.

Similarly, when we performed transduction from lysates generated from the SH1000 80 α SaPI1 Δ *cpmAB* pGO1 background, we obtained three pGO1 variants from two of the three independent biological repeats: pGO1_{evolE} (replicate 1), no transductants (replicate 2), and pGO1_{evolH} and pGO1_{evolI} (replicate 3).

2.16 I'm curious about the evolved pGO1evolF and pGO1evolA, that show differences in transfer efficiencies when in competition. Do you have any hypothesis for the reason behind it? Because in the case of pGO1evolA is practically the same as pGO1evolB as far as I saw.

This is a very nice observation, that unfortunately, we can't answer properly. We can argue that any small difference in sequence may have a high impact either on conjugation or transduction. But we have not demonstrated that for the examples mentioned by the reviewer.

2.17 Line 250/262 Plasmid genome size evolution /recombination due to the presence of IS elements – (described as fascinating) – the contribution of IS elements to plasmid genome rearrangements has been studied (and published) quite extensively in the 1970s (e.g., see this review: <https://www.nature.com/articles/263731a0>).

We have rewritten part of this section to reflect the reviewer's point. Lines 290-292 now read: *"These data are consistent with others' observations that IS elements facilitate distinct remodelling strategies for the evolution of medium-to-large composite plasmids²⁷."*

2.18 Line 332 – the title of this section should be toned down. ... transfer occurs in milk (not nature). While the experiment in milk is a nice addition to test the boundaries of plasmid-mediated transfer, I think that also here, the conclusions go far beyond the clear observations.

We have amended the title of this section to read: *"Phage- and PICI-mediated plasmid transfer occurs in milk."*

We have also amended line 349 to read: *"Accordingly, milk represents a more natural model environment for examining phage- or PICI-mediated plasmid transfer."*

2.19 Line 366 – The context of the statistical test is not clear. Looking at Fig. 7, about half of the isolates with SaPIs harbor also small plasmids (50/95), which would be then close to random (i.e., flip of a coin).

We thank the reviewer for this comment. It would be close to random if half of the total *S. aureus* genomes carried small plasmids, which is not the case (only 68/295=30%), hence the significance of the association between SaPIs and small plasmids.

2.20 Line 371 – the analysis comparing plasmid and phage sizes is extremely superficial and the conclusions far-fetched. Seeing two distributions with a similar shape cannot be used to conclude about causality (in other words, beware of the storks! A new parameter for sex education | Nature <https://www.nature.com/articles/332495a0>)

We agree with the reviewer, and we have attempted to tone down our conclusions from this section by using more suggestive language throughout, as well as by adding the following caveat at the end of the section:

(Lines 401-404) *"However, other factors apart from phage-mediated mobilization, such as for example differences in host defense systems, may help to explain the co-occurrence of phages and plasmids in genomes, and further work will be required to confirm our hypothesis."*

Discussion

2.21 In the databases surveyed, the plasmids you found, where there traces in their genomes of phage and PICIs sequences? Because that could serve as support for these plasmids found

were maybe reduced in size due to transduction capsid size limitations, like in the case of pGO1 hybrids and possibly pACK2.

We thank the reviewer for this great suggestion. Actually, there was a small but significant number of plasmids sequences that contained phage genes in them and were also detected as phages. In our initial analysis we filtered those out to avoid potential confounding effects. However, we will look into them more carefully in a future project that we are now starting.

2.22 Lines 427-429 – the authors demonstrate this statement (on mobilizable plasmids) only for *S. aureus* and their conclusions should be state accordingly. They can “suggest” that their finding is more general.

We have amended this section to make it more specific to *S. aureus*. The section covered in lines 443-446 now reads:

“Our results demonstrate that in S. aureus, mobilisable plasmids can significantly increase their transferability by exploiting phages, PICIs and conjugative plasmid machineries for transfer...”

2.23 Line 430 – previous studies demonstrated plasmid transfer by phages, also specifically for *S. aureus* – Im quite surprised that studies from R. Pantůček on the topic are not cited in this manuscript.

We did not intend to imply that plasmid transfer via phages has not been previously reported, but that specifically, our results demonstrate that plasmid transfer can be facilitated by phages (and/or SaPIs) between different bacterial species and genera. We have amended this statement to attempt to clarify the point that we intended to make. The statement now reads (line 447):

“Our results also demonstrate that interspecies and intergeneric mobilisation of plasmids is not exclusively mediated by conjugation but may also occur via transduction.”

We thank the reviewer for reminding us of the work of R. Pantůček. We have added a relevant citation into the following sentence on lines 456-457:

“Our results add to the recently recognized concept of “silent transfer” of pathogenicity and antibiotic resistance factors carried by MGEs^{26,29,30,46} by phages that cannot grow on the target organism”.

2.24 Line 438 – about the consequences of phage host range to horizontal transfer by transduction, see this publication, based on phylogenetic reconstruction: doi:10.1038/ismej.2016.116

We thank the reviewer for providing this reference. We have added the following sentence in lines 457-462:

“It should be noted, however, that while phylogenetic analysis has indicated that intra- and intergeneric phage-mediated transfer events do occur in natural bacterial populations, they do so at extremely low frequencies⁴⁷, suggesting that while the potential for phage and/or PICIs to contribute to plasmid transfer within polymicrobial communities exists, it is unlikely to occur at significant rates.”

2.25 Line 468 – on phages shaping plasmid evolution – this is an exaggerated statement. The findings are specific for *S. aureus* and the effect suggested by the results is likely on plasmid genome size (not, e.g., gene content).

We have amended this section to make it more specific to *S. aureus* and to clarify that the effect of phage/SaPI selection pressure on plasmid evolution is due to its impact on genome size. The section covered in lines 490-495 now reads:

“Our results have also demonstrated that phages and SaPIs not just promote plasmid transfer, but may also contribute to plasmid evolution in S. aureus, by selecting for variant plasmids that have undergone IS-mediated remodelling, as has previously been described for mosaic-like plasmids in S. aureus²⁷, to sizes compatible with packaging into either in the phage or SaPI-sized capsid, with implications for plasmid gene content due to reduced genome size”.

2.26 In line 509-512– again, an exaggerated conclusion – considering that phages are not universal for all taxa. The authors do not have results to support that conclusion. Also, I do not fully agree that less genes decreases the vertical transmission. If a plasmid has addiction system it will be transmitted regardless of the number of genes it carries. It will only be less fit according to the host point of view in presence of selection. In other words – I think that the last discussion paragraph is very speculative (of course, it’s up to the authors to write their opinion, yes, a more suggestive language would be prudent).

We have amended this section to be more specific to *S. aureus* and use more suggestive language. The section covered in lines 520-524 now reads:

“Overall, we propose that in highly lysogenic species such as S. aureus, horizontal transfer via transduction is favoured for plasmids where there is a high benefit to the plasmid in being small. In some circumstances, however, the benefit conferred to the host cell by the plasmid may be reduced owing to fewer genes being carried, thus potentially reducing plasmid fitness in terms of vertical transmission.”

Reviewer #3 (Remarks to the Author):

We thank the reviewer for his/her nice comments about the manuscript.

Minor comments:

3.1- Line 110: reference here your supplementary table with the accession numbers (ST7). You do not mention how you selected the 295 genomes. A quick search on NCBI reveals that there are many more (including complete ones).

The reviewer is correct. The number of genomes is much higher now, but we performed this analysis more than two years ago (15/02/2019). We performed the bioinformatic analyses prior to the experimental ones to build our hypothesis, and that was the number available back then. We performed all the analysis on the genomes available at that time, as specified in the methods section (line 674). Given the accelerating rate of whole genome sequences deposited in databases the number has significantly increased since then.

3.2- Are multiple copies of pC221 present in the SaPI1 capsids? Or are those small SaPI-sized capsids stable and can accommodate smaller genomes (4.6 kbp vs. 15 kbp)?

We propose that this would be dependent on the mechanism of replication used by the plasmid. In the case of pC221, this plasmid replicates via the rolling circle mechanism commonly utilised by small plasmids (DOI: 10.1111/j.1365-2958.1993.tb01625.x), similar to the concatemeric replication via RC observed for both phages and SaPIs. We propose that due to the RC mechanism of pC221 DNA replication, each transducing SaPI-sized capsid would contain three copies of linear pC221 DNA, with packaging initiating from the *ppac* site on the plasmid and terminating when the capsid capacity is reached (~15 kb), thus ensuring capsid stability during maturation and transduction. Recombination of the linear plasmid DNA within the host cell following injection from the transducing particle would enable circularisation of the complete plasmid in the host cell, as would be the case following pC221 mobilisation via a conjugative plasmid, such as pGO1.

3.3- Although I do think the conclusions are otherwise well supported, it appears that you did not treat your lysates with DNase (or at least that is not mentioned in your M&M, only 0.2 µm size filtration is mentioned). That must mean that abundant amounts of DNA is available outside of the capsids when conducting your main transduction assays to *S. aureus*, *S. epidermidis*, *S. xylosum*, and *L. monocytogenes*. *S. aureus* does have competence mechanism via SigH, and it could be argued that limited amounts of transformation could occur and confound your results – could you clarify whether you think that could be the case or if you ruled that out completely?

- Furthermore, Mitomycin C is also still present in the lysate that you use to transduce your cells. In some species, it has been shown to induce competence.

The reviewer raises an interesting possibility, and we thank them for their insight on this topic, however we do not feel that natural transformation with exogenous DNA is likely to be a confounding factor for what we observed in our results. While a putative competence mechanism operating via SigH has been described for *S. aureus*, there is extremely limited evidence of natural transformation occurring under laboratory conditions, with no evidence of SigH-mediated expression observed for *S. aureus* grown in standard culture media under standard laboratory conditions (Morikawa et al (2012), <https://doi.org/10.1371/journal.ppat.1003003>) such as those used in this study. Furthermore, we consistently did not observe any transductants from any experiment where the 80αΔ*terS*/80αΔ*terS* SaPI1Δ*terS* mutants were used as donor strains. Given that similar levels of exogenous DNA and residual MC would be expected to be present in these lysates to that

in the other donor lysates, this suggests that plasmid transfer required the presence of active phage/SaPI particles and so resulted from GT, not natural transformation.

3.4- Could you explain further how the five evolved plasmids (pGO1evol_a to f) were created? You mention multiple separate experiments but it isn't clear why you would need multiple experiments... or do you mean the replicates of the experiments that led to Figure 1C?

As explained to Reviewer 2 in comment 15, early on in this work we attempted to generate evolved mutants of pGO1 via transduction with phage 80 α . Following many attempts, three transductants were obtained, namely pGO1evol₁, pGO1evol₂ and pGO1evol₃ (all of which were identical in terms of rearrangement so only pGO1evol was included in this manuscript). Serendipitously, when we performed the initial transduction pilot experiments using the 80 α SaPIbov2 pGO1 background strain, we unexpectedly obtained four transductants across three biological replicates: no transductants were obtained from replicate 1, one transductant - pGO1evol_A - was obtained from replicate 2, and three transductants - pGO1evol_B, pGO1evol_{B2} and pGO1evol_{B3} (all were identical, so only pGO1evol_B was described in this manuscript) - were obtained from replicate 3.

Following staffing changes in the project, the parental donor strain SH1000 pGO1 80 α SaPIbov2 was remade (strain JP19996) and the experiment was performed again to ensure that the transductants obtained in the initial pilot experiments were not an artefact. The results of the repeated experiment are those displayed in Fig. 1C and gave rise to plasmid/SaPIbov2-plasmid variants: pGO1evol_C and pGO1evol_F (replicate 2), and pGO1evol_J (replicate 3). In addition, in order to confirm that the unusually large capsid size of SaPIbov2 permitted selection of remodelled pGO1 variants, we also performed some induction/transduction experiments using SH1000 80 α SaPI1 Δ cpmAB pGO1 (since this strain does not produce small capsids, similarly to SaPIbov2). From these three repeats we obtained: pGO1evol_E (replicate 1), no transductants (replicate 2), and pGO1evol_H and pGO1evol_I (replicate 3). The fact that we obtained transductants harbouring remodelled pGO1 variants from the remade JP19996 donor strain validated the results that had been obtained from the initial experiments, and since the plasmids produced from the earliest experiments yielded different pGO1 variations from those obtained in the subsequent experiments (which led to Fig. 1C), we felt that they warranted further investigation, resulting in their inclusion in the manuscript despite not being part of the data presented in Fig. 1C.

3.5 - In line 372, do you mean any plasmids? Or the non-transmissible plasmids?

The analysis in this section includes all plasmids, not just the non-transmissible plasmids.

3.6 - In your discussion (431-435) you warn about the impacts of the mechanism you have described in the field of phage therapy. This is very interesting, but as far as I know, we do not use "poorly transducing" phages for therapy - typically, if the genome analysis shows a sign of potential lysogenic lifestyle, the phage is no more a candidate. Do you contend that strictly lytic phages could also be hijacked for the transduction of plasmids as you have described here?

Lytic phages can engage in GT since this mechanism is driven by the misrecognition of *ppac* sites on plasmids or the bacterial chromosome by the phage TerS. Furthermore, cell infection (in the lytic cycle) by a helper phage is sufficient to induce SaPI replication and packaging, thus leading to the initiation of SaPI-mediated GT of plasmids. Accordingly, we have removed the words 'and poorly transducing' from this sentence in line 449. The sentence now reads:

"As bacterial pathogens become increasingly antibiotic resistant, lytic phages have been proposed for phage therapy..."

REVIEWERS' COMMENTS

Reviewer #1 (Remarks to the Author):

The authors have done an excellent job revising their manuscript. They made textual changes to tone down some of their statements. Where experiments/bioinformatics analyses were requested, the authors either carried out the experiments as requested, or explain why the suggested experiments are not feasible. To better understand the interaction between prophage and plasmid carriage (my comment 1.7) the authors have attempted an analysis, which unfortunately yielded inconclusive results. I don't think the proposed analysis is essential for this paper, and I hope the authors will be able to address this point in future studies.

However, I have one final (and very easy to address) point - the response to my point 1.5 is very clear and convincing - given the points made about overlapping genes, the overexpression strategy employed makes a lot more sense. It would make sense, however, to generate a graph of these data and include these into the manuscript.

Apart from this, I am satisfied with the changes made by the authors in response to my comments and suggestions.

Reviewer #2 (Remarks to the Author):

The authors replied all of my comments/concerns satisfactorily. I have no doubt that other scientists from diverse fields (e.g., microbial evolution/ecology/genetics) will find it interesting and fun to read.

Reviewer #3 (Remarks to the Author):

The authors have answered all my previous comments satisfactorily.

REVIEWER COMMENTS

Reviewer #1 (Remarks to the Author):

The authors have done an excellent job revising their manuscript. They made textual changes to tone down some of their statements. Where experiments/bioinformatics analyses were requested, the authors either carried out the experiments as requested, or explain why the suggested experiments are not feasible. To better understand the interaction between prophage and plasmid carriage (my comment 1.7) the authors have attempted an analysis, which unfortunately yielded inconclusive results. I don't think the proposed analysis is essential for this paper, and I hope the authors will be able to address this point in future studies.

However, I have one final (and very easy to address) point - the response to my point 1.5 is very clear and convincing - given the points made about overlapping genes, the overexpression strategy employed makes a lot more sense. It would make sense, however, to generate a graph of these data and include these into the manuscript.

Following the reviewer's comments, we have now included a paragraph in the manuscript describing these data.

Apart from this, I am satisfied with the changes made by the authors in response to my comments and suggestions.

Reviewer #2 (Remarks to the Author):

The authors replied all of my comments/concerns satisfactorily. I have no doubt that other scientists from diverse fields (e.g., microbial evolution/ecology/genetics) will find it interesting and fun to read.

Reviewer #3 (Remarks to the Author):

The authors have answered all my previous comments satisfactorily.